

Quantifying Temperature-sliding Inconsistency in Thermomechanical Coupling: A
Comparative Analysis of Geothermal Heat Flux Datasets at Totten Glacier
Junshun Wang[1], Liyun Zhao[1], Michael Wolovick[2], John C. Moore[3]
[1]State Key Laboratory of Earth Surface Processes and Hazards Risk Governance
(ESPHR), Faculty of Geographical Science, Beijing Normal University, Beijing
100875, China
[2]Glaciology Section, Alfred-Wegener-Institut, Helmholtz-Zentrum für Polar- und
Meeresforschung, Bremerhaven, Germany
[3]Arctic Centre, University of Lapland, Rovaniemi, Finland
*Correspondence: Liyun Zhao (zhaoliyun@bnu.edu.cn), John C. Moore*
*(john.moore.bnu@gmail.com)*

**Abstract.** Rapid sliding of ice sheets requires warm basal temperatures and lubricating
basal meltwater, whereas slow velocities typically correlate with a frozen bed. However,
ice sheet models often infer basal sliding by inverting surface velocity observations
with the vertical structure of temperature and hence rheology held constant. If the
inversion is allowed to freely vary sliding over the model domain, then inconsistencies
between the basal thermal state and ice motion can arise lowering simulation realism.
In this study, we propose a new method that quantifies inconsistencies when inferring
warm and cold-bedded regions of ice sheets. This method can be used to evaluate the
quality of ice sheet simulation results without requiring any englacial or subglacial
measurements. We apply the method to evaluate simulation results for Totten Glacier
using an isotropic 3D full-Stokes ice sheet model with eight geothermal heat flux (GHF)
datasets and compare our evaluation results with inferences on basal thermal state from
radar specularity. The rankings of GHF datasets based on inconsistency are closely
aligned with those using the independent specularity content data. Examples of the
method utility are 1. an inconsistency characterizing overcooling with all GHFs near
the western boundary of Totten Glacier between 70°S-72°S, where there is a bedrock
canyon and fast surface ice velocities, which suggests that GHF is low in all published
datasets; 2. an overheating inconsistency in the eastern Totten Glacier with all GHFs
that leads to overestimation of ice temperature due, in this case, to an unrealistically
warm surface temperature. Our approach opens a new avenue for assessing the self-
consistency and reliability of ice sheet model results and GHF datasets, which may be
widely applicable.



## 1. Introduction

Ice sheet models are an important tool for projections of ice sheet mass balance and their contribution to sea level rise. Ice sheet models are usually initialized by "spin-up" or data assimilation such that they reproduce the present-day geometry or surface velocity of an ice sheet (Seroussi et al., 2019). Often ice sheet model simulations derive ice dynamics using ice temperatures taken from other studies (e.g., Gillet-Chaulet et al., 2012; Van Liefferinge and Pattyn, 2013; Cornford et al., 2015; Pittard et al., 2016; Siahaan et al., 2022). In thermo-mechanically coupled ice sheet simulations, the ice sheet model is usually spun up with idealized temperature-depth profiles and then run in a thermo-mechanically coupled mode constrained by geothermal heat flux (GHF) and surface ice temperature fields (Seroussi et al., 2019). While advances in satellite and field observation technologies have led to a preliminary consensus on ice sheet geometry and surface ice temperature, significant uncertainties persist in basal boundary conditions, including GHF and basal friction, since reliable observational data are scarce. These basal properties introduce significant uncertainty in the simulated ice sheet dynamics, and thus ice sheet mass balance.

The GHF, the heat flow from the Earth's crust to the base of ice sheet, is a critical variable in the basal boundary condition for simulating the ice temperature profile, and hence ice rheology and flow dynamics (Fisher et al., 2015; Smith‑Johnsen et al., 2020; Reading et al., 2022). Several GHF datasets exist, derived in various ways from geophysical observations and models, and they exhibit significant variability in both spatial distribution and magnitude (e.g., An et al., 2015; Dziadek et al., 2017; Martos et al., 2017; Shen et al., 2020; Stål et al., 2021). These GHF datasets have been widely used in thermodynamic simulations of Antarctica (e.g., McCormack et al., 2022; Shackleton et al., 2023; Park et al., 2024; Van Liefferinge et al., 2018). However, assessing the GHF field accuracy is problematic because in situ measurements such as boreholes are sparse. Few studies have assessed the quality and reliability of GHF datasets over specific regions. Kang et al. (2022) employed a combination of forward model and inversion using a 3D full-Stokes ice flow model to simulate the basal thermal state in the Lambert–Amery Glacier region and evaluate different GHFs using the locations of subglacial lakes, but the constraints used were asymmetric between cold and warm beds, and assigned inflated reliability to the warmer GHF maps. Indirect estimates of basal conditions have used airborne radar specularity content (Schroeder et al., 2013, 2015; Young et al., 2016) as proxies for basal wetness/dryness and thermal regime (Dow et al., 2020). Huang et al. (2024) used an inverse modeling approach similar to that of Kang et al. (2022) for Totten Glacier and combined this with measured radar specularity content to derive a two-sided constraint on the basal thermal state in addition to subglacial lakes locations. However, specularity content is not yet available for many regions of Antarctica.





The basal friction field is another poorly known boundary condition in ice sheet
modeling, and a key source of uncertainty in the long-term projection of ice sheets and
glaciers. Although basal slip is crucial to the 3D ice flow, it is difficult to observe.
Several basal sliding parameterizations have been proposed and widely used
(Weertman, 1957; Kamb, 1970; Nye, 1970; Budd et al., 1979; Fowler, 1981; Schoof,
2005; Gagliardini et al., 2007; Gladstone et al., 2014; Tsai et al., 2015; Brondex et al.,
2017, 2019). The linear Weertman basal sliding parameterization is the most widely
used due to its simple form. Given prescribed or modelled ice temperatures and hence
ice viscosity, numerous studies have inferred the spatial distribution of basal friction
coefficient over grounded ice to best match observed present-day surface ice velocities
or ice sheet geometry using snapshot or time-dependent data assimilation and inverse
methods (MacAyeal, 1993; Morlighem et al., 2010; Rignot et al., 2011; Gillet-Chaulet
et al., 2012; Larour et al., 2012; Pollard and DeConto, 2012; Morlighem et al., 2013;
Perego et al., 2014; Pattyn, 2017; Albrecht et al., 2020; Lipscomb et al., 2021; Choi et
al., 2023). However, such inversions typically allow the friction coefficient to vary
freely to match the surface velocity observations. This can potentially lead to conflicts
with the temperature field used during the inversion, which we refer to as
"inconsistencies" in this study. For instance, relatively fast surface ice velocity may
demand basal sliding in areas where the basal temperatures are below the local pressure
melting point. These inconsistencies may be due to unrealistic ice temperatures or a
lack of complete physics in the ice sheet model. However, many studies overlook this
aspect, and use the inversion results to initialize ice sheet dynamics simulations and
estimate glacier mass balance and its contribution to sea level rise (Seroussi et al., 2019;
Peyaud et al., 2020; Schannwell et al., 2020; Payne et al., 2021).
To the best of our knowledge, there has been no study of such inconsistencies
between simulated ice temperature and observed surface ice velocity. Here we develop
a novel and generally applicable method to estimate this inconsistency without relying
on basal observation data. We utilize the inconsistency of the modelled ice temperature
and observed velocity fields to evaluate the quality of ice flow model results. Notably,
this approach can also serve as a supplementary method for assessing geothermal heat
flux datasets, relying solely on surface ice velocity observations rather than additional
englacial or subglacial data.
We apply our method to Totten Glacier, a primary outlet of the Aurora subglacial
basin in East Antarctica (Greenbaum et al., 2015; Pritchard et al., 2009). The Totten
Glacier subregion experienced the largest mass loss among drainage basins in East
Antarctica during the period 1979-2017 and 2003-2020 (Kim et al., 2024; Rignot et al.,
2019) (Fig. 1a). We examine inconsistencies between simulated ice temperature and ice
velocity fields using a 3D full-Stokes model using the various GHFs included in Huang
et al. (2024) and use this analysis to rank the reliability of different GHF fields. This





GHF ranking closely resembles that reported by Huang et al. (2024), which used the
agreement between the modelled basal thermal regime and specularity content, which
we take as a validation of the method. Since the new method does not require any
englacial or subglacial data, it can be applied to many glaciers, particularly those
lacking observations. Our approach can provide a swift assessment of the plausibility
of basal temperature and velocity simulated by ice sheet models. Additionally, it can be
effectively utilized to map the spatial distribution of GHF over- or under-estimation.
**2. Method**
The inconsistencies defined in this study are essentially between the modelled
basal thermal state and observed surface ice flow motion. More specifically, the
inconsistencies are between modelled frozen bed and modelled basal sliding (which is
tuned to match the observed fast surface velocity during the inversion), and between
modelled warm bed and observed slow surface velocity. The inconsistencies originate
from multiple causes, including uncertainties in GHF, surface ice temperature, ice sheet
geometry, bed topography, surface velocity, ice density and incomplete ice flow
mechanics.
There is no direct correlation between basal temperature and surface velocity;
rather, they are linked through the basal thermal state - the basal temperature being at
or below the pressure melting point. The ice bottom in the study domain can be
partitioned into warm and cold beds depending on whether the simulated basal ice
temperature reaches the local pressure melting point. To effectively penalize models
exhibiting both localized overheating (bed too warm) and overcooling (bed too cold),
we establish overheating metrics within the warm-bedded region and overcooling
metrics within the cold-bedded region to quantitatively assess the inconsistency
between the simulated temperature and velocity fields. Thus, we provide two-sided
constraints on the temperature field that penalize both too high and too low ice
temperature.
Overcooling occurs where basal temperature is underestimated. Crucially, in
regions with relatively fast observed surface velocity, the inverse method nevertheless
yields a nonzero basal velocity — a physically inconsistent result given the cold basal
temperature. When basal ice temperature is below the pressure melting point, the basal
modelled velocity is expected to approach zero. This inconsistency is larger for faster
simulated basal ice speed and for colder simulated basal temperatures. We therefore use
a formula that accounts for both variables to quantify overcooling:
$$AOC = (T_{melt} - T_{bm}) \times U_{bm}, \qquad (1)$$
where $AOC$ stands for absolute overcooling, $T_{melt}$ is the basal pressure melting point,
$T_{bm}$ represents the simulated basal ice temperature and $U_{bm}$ means the simulated basal
ice speed.



For the overheating metric, since the first term of the right-hand side of Eq. (1)
becomes zero at a warm bed, we cannot use a similar formula as Eq. (1). It is not
straightforward to quantify the inconsistencies between modelled warm bed and
expected slow basal speed given slow observed surface speed. We note the fact that
modelled basal sliding speed must remain non-negative. If the ice is warm and soft
enough to permit deformation such that the modelled surface speed is much faster than
the observed, then a friction inversion will be ineffective to correct this misfit,
producing a bias towards positive misfits (i.e., model velocities are too fast) in the
inversion results. Therefore, we use the positive difference between the simulated
surface ice speed and the observed speed to calculate the inconsistency caused by the
overheating effect:
$$AOH = \max(0, U_{sm} - U_{obs}), \quad (2)$$
where $AOH$ refers to absolute overheating, $U_{sm}$ represents the modelled surface ice
speed and $U_{obs}$ is the observed surface ice speed. We only calculated $AOH$ for the warm-
bedded areas, i.e. $T_{bm} = T_{melt}$, because observed surface ice speed errors are
proportionally much less in warm-bedded areas (corresponding to fast flow regions)
than in cold-bedded area (correspond to slow flow regions).
To mitigate the impact of substantial differences in observed surface ice speed
across various areas, we also define "relative overheating" ($ROH$) and "relative
overcooling" ($ROC$), dividing $AOH$ and $AOC$ by the observed surface ice speed
respectively:
$$ROH = \frac{\max(0, U_{sm} - U_{obs})}{U_{obs}}, \quad (3)$$
$$ROC = (T_{melt} - T_{bm}) \times \frac{U_{bm}}{U_{obs}}. \quad (4)$$
The summation of the above four metric values is computed across grid points
where each metric is explicitly defined. Specifically, $AOH$ and $ROH$ metrics are
computed over the warm bed region, and $AOC$ and $ROC$ metrics are computed over the
cold bed region for each simulation result. This summation approach was chosen to
preserve the total magnitude of inconsistencies, as the warm bed and cold bed regions
are different due to distinct GHF boundary conditions. Furthermore, since all
experiments utilize identical mesh, the cumulative values remain directly comparable
for cross-experiment analysis. We only consider grounded ice and exclude points
located at the domain boundary due to relatively poor model performance there.
To evaluate the inconsistencies for the whole domain, we linearly normalized the
overheating inconsistency and overcooling inconsistency to range from 0 to 1 and then
sum them as:
$$ACI = L_N(AOC) + L_N(AOH), \quad (5)$$
$$RCI = L_N(ROC) + L_N(ROH), \quad (6)$$





where *ACI* means absolute combined inconsistency, *RCI* represents relative combined
inconsistency, and $L_N$ represents linear normalization. Taking *AOC* as an example, its
linear normalization is:

$$L_N(AOC) = \frac{AOC - AOC_{min}}{AOC_{max} - AOC_{min}}. \qquad (7)$$

where $AOC_{min}$ and $AOC_{max}$ denote the minimal and maximal *AOC* values across all the
simulation results when multiple simulation outcomes are available. Therefore, we
obtain 6 metrics consisting of three absolute inconsistencies *(AOH, AOC, ACI)* and
three relative inconsistencies (*ROH, ROC, RCI*).
These 6 indicators can comprehensively analyze the temperature-sliding
inconsistency in the inversion results of ice sheet model. For each metric, simulation
results are assigned ranks ranging from 1 to *N* (where *N* represents the total number of
simulation results), with 1 indicating the smallest inconsistency and *N* the largest. The
final score for each simulation result is subsequently calculated as the arithmetic mean
of its six metric-derived scores, ensuring a comprehensive evaluation framework. as a
reasonable simulation result should perform well across warm bed, cold bed, and the
whole region.

**3.  Application to Totten Glacier with Different GHFs**
**3.1  Study domain and Data**
We apply our method to evaluate simulated ice temperature and ice velocity in
Totten Glacier by following Huang et al. (2024) and using eight GHF datasets. Huang
et al. (2024) used the present-day surface ice temperature (Le Brocq et al., 2010) and
ice sheet topography data from BedMachine Antarctica, version 2 (Morlighem et al.,
2020). The eight GHF datasets were derived by various methodologies, resulting in
significant differences in both spatial distribution and magnitude (Fig. 2). GHF fields
from Stål et al. (2021), Haeger et al. (2022), Lösing and Ebbing (2021) and Martos et
al. (2017) generally exhibit higher magnitudes than the other GHFs.

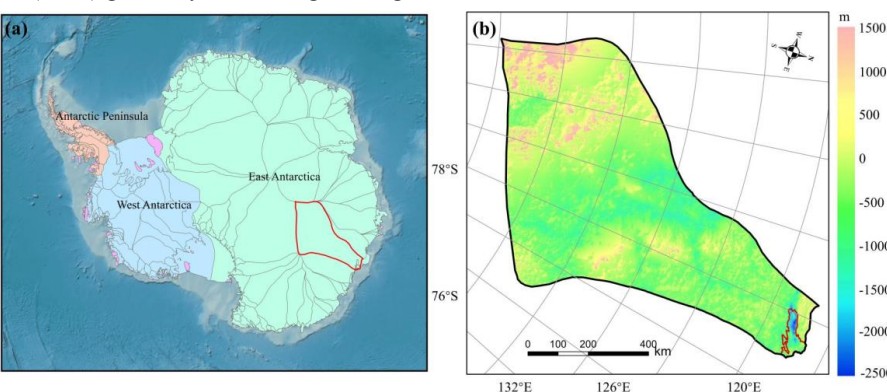






**Figure 1. (a)** Geographic location of Totten Glacier (red outline) in Antarctica; **(b)** bed elevation of Totten Glacier, the red curve represents the grounding line.

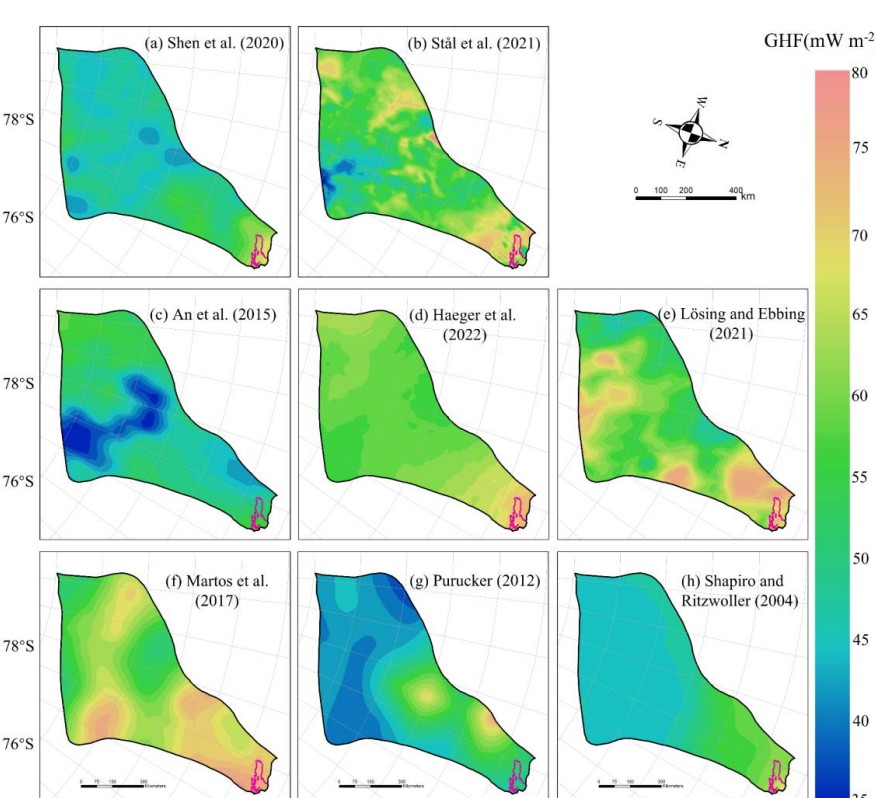

**Figure 2.** The spatial distribution of the 8 GHF datasets for Totten Glacier **(a–h)** used as input data in Huang et al. (2024). The purple line depicts the grounding line.

The spatial distribution of modelled basal temperature using the 8 GHFs displays both similarities and heterogeneity. In the northern part of Totten Glacier, there is a consistent warm-bedded pattern across all eight simulation results (Fig. S1), which originates from the grounding line and extends upstream to approximately 71°S. This warm-bedded area is not contiguous with the lateral boundaries of Totten Glacier but is instead bordered by cold bed. All 8 GHF datasets produce low basal ice temperatures in the inland southwest, with Purucker et al. (2012), Shapiro and Ritzwoller (2004), Shen et al. (2020) and Lösing and Ebbing (2021) being colder. The basal ice velocities





modelled from the 8 different GHF datasets produce similar spatial distributions (Fig.
S2), which can be expected as they were derived using the same inverse method and
constrained by the identical observed surface ice velocity. The modelled basal ice
velocity is fast near the grounding line and its upstream area. There are also high
velocities between 70°S and 72°S close to the western boundary of Totten Glacier,
which are associated with subglacial canyon features in the basal topography (Fig. 1b)
and observed fast surface ice velocity there.

**3.2  Spatial Distribution of Inconsistencies**
We calculate the absolute inconsistencies, *AOH*, in the warm bed, and *AOC* in the
cold bed. The spatial distribution of *AOC* reveals that most GHF datasets exhibit
significant local overcooling inconsistencies at the subglacial canyon between 70°S and
72°S (Fig. 3). There is fast basal sliding in the inverse model results (Fig. S2), however,
the modelled basal ice temperatures inferred from most of the GHF datasets are below
the pressure melting point (Fig. S1). High specularity content in radar data (Fig. 3c)
suggests the presence of basal water in the subglacial canyons here (Dow et al., 2020;
Huang et al., 2024), which also suggests that the basal ice temperature should be at the
pressure melting point and confirms the inconsistency between the modelled
temperature and velocity fields.
The area near the grounding line is characterized by fast ice flow and warm bed
(Fig. 3), yet some of the margin is cold-bedded with modelled basal temperature below
the pressure melting point, resulting in high *AOC*. Overall, modelled results with most
GHF datasets show small overcooling inconsistencies. The modelled results using GHF
from Purucker et al. (2012), Shapiro and Ritzwoller (2004), Shen et al. (2020), Lösing
and Ebbing (2021) exhibit no overcooling inconsistency in southwestern Totten Glacier
(Fig. 3).
The spatial distribution of relative overcooling inconsistencies, *ROC* (Fig. 4),
differs from that of absolute inconsistencies, *AOC*, and is due to the spatial variability
in surface ice speed. The largest value of *ROC* across most GHF occurs at Dome C,
where the observed surface ice speed is close to zero.

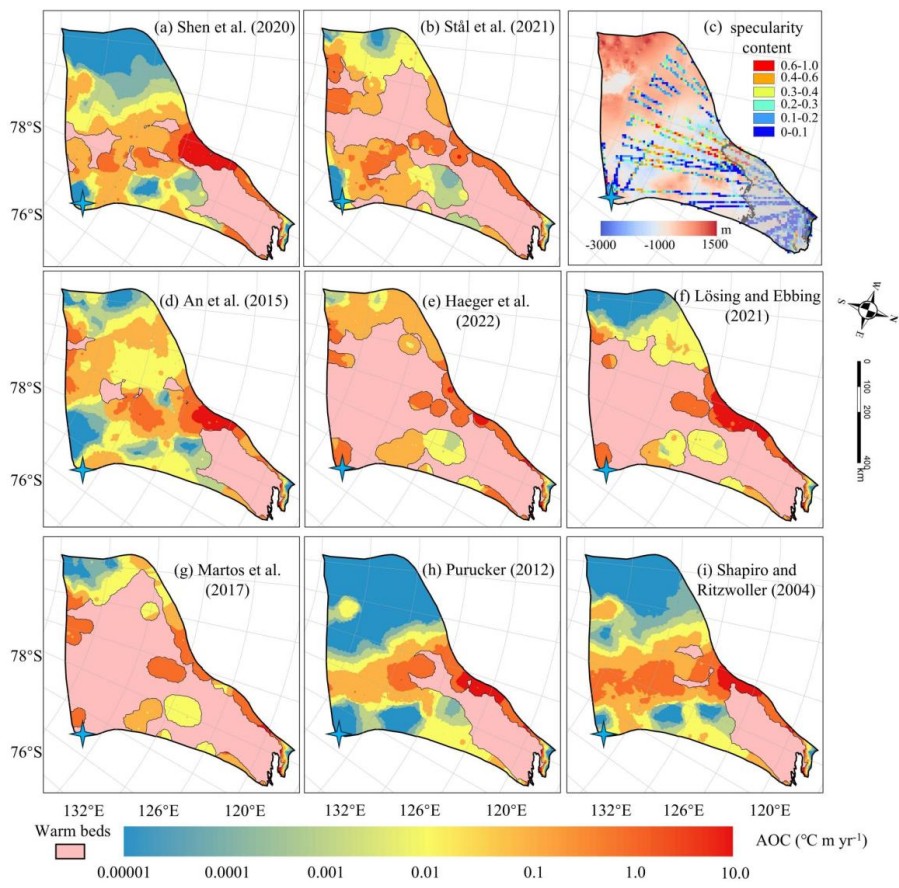

**Figure 3.** Spatial distribution of *AOC* inconsistency in modelled cold-bedded regions **(a-b, d-i)** associated with the GHFs **(a–h)** in Fig. 2. The colormap is on logarithmic scale. The pink region represents modelled warm bed. **(c)** Specularity content sourced from radar data collected by ICECAP (Dow et al., 2020) with the bed elevation in the background. Gray area in **(c)** corresponds to surface speed exceeding 30 m yr⁻¹. The blue star represents Dome C.



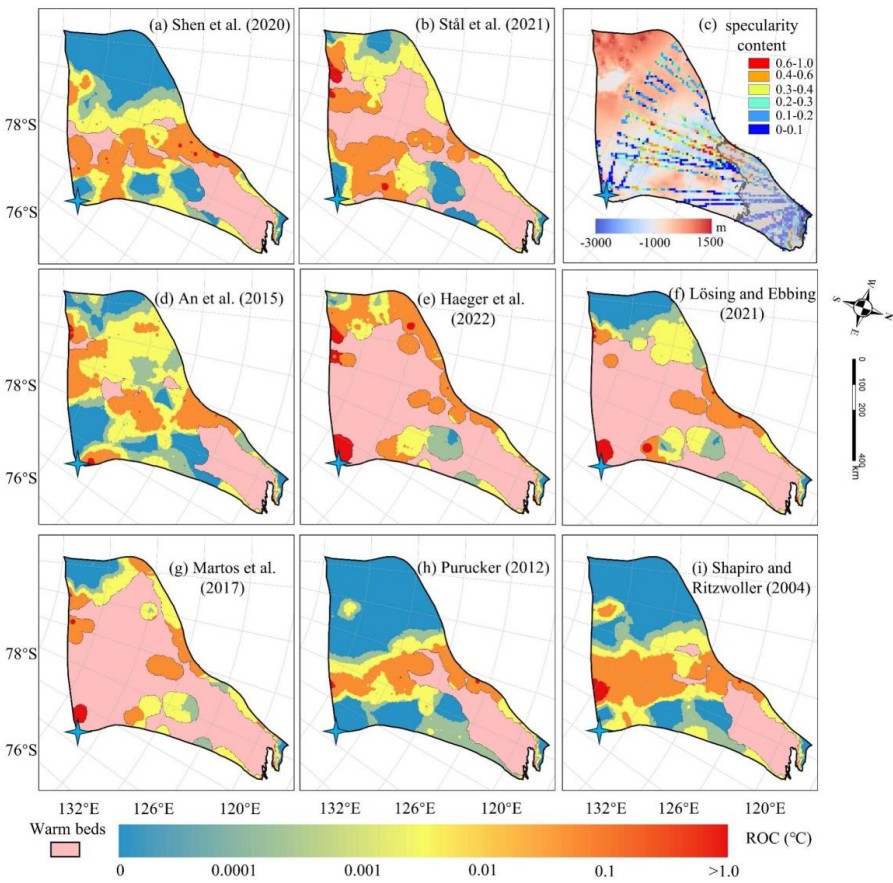

**Figure 4.** The spatial distribution of relative overcooling (*ROC*) inconsistency in cool beds with (a), (b) and (d) to (i) corresponding to the GHFs (a – h) in Figure 2. The pink area represents the warm beds. Dome C is marked by a blue star. (c) Locations of specularity content derived from radar data collected by ICECAP (Dow et al., 2020) and with the bed elevation in the background. The gray curve is the contour of the surface speed of 30 m yr$^{-1}$. Note the colormap is non-linear.

The GHF datasets of Stål et al. (2021), Haeger et al. (2022), Lösing and Ebbing (2021) and Martos et al. (2017) which have higher than average GHF values provide larger areas of warm bed than the other 4 GHFs. The simulations with all 8 GHFs yield similar spatial distributions of *AOH* (Fig. 5) on the common area of warm bed, and similar locations of high *AOH* values. A common high *AOH* area is located between 69°S and 72°S in the eastern part of Totten Glacier, due to simulated surface ice





velocities greatly exceeding the observed surface ice velocities. Low specularity content from radar data (Fig. 5c) suggests there is no basal water in the area (Dow et al., 2020; Huang et al., 2024). Therefore, it is likely that the basal ice temperature is overestimated there. The simulations with all the 8 GHFs also yield similar spatial distribution of *ROH* (Fig. 6), but its largest values are mostly in the slow flowing region as one may expect from its formulation (Eq. (3)).

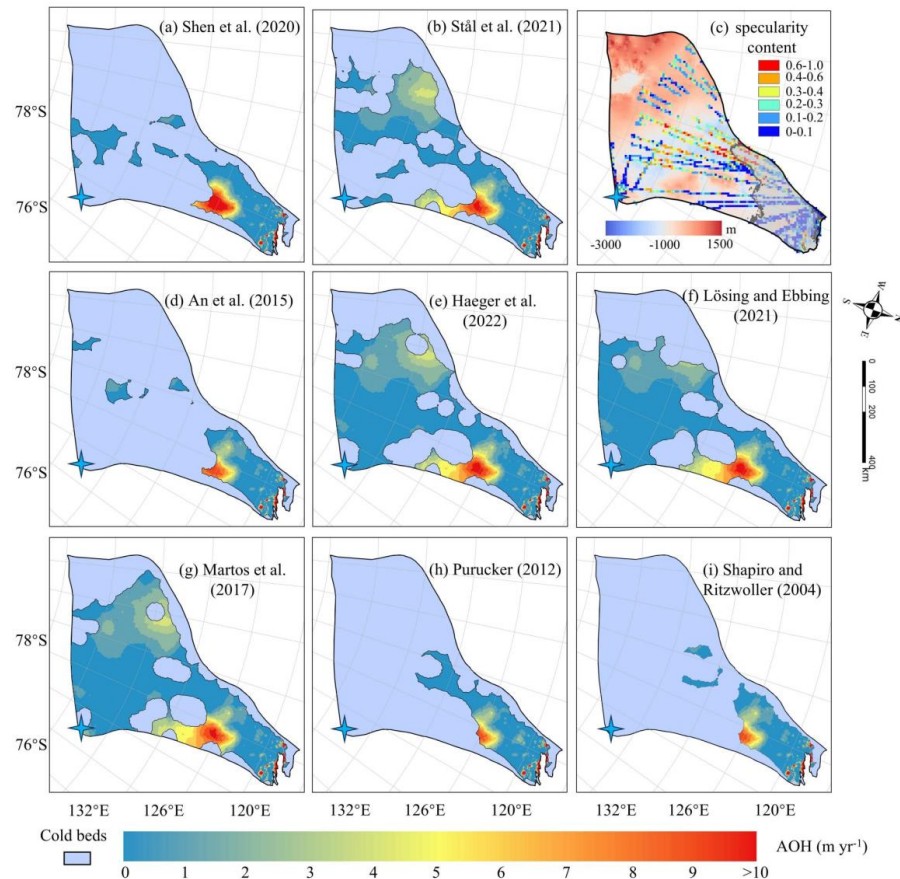

**Figure 5.** Spatial distribution of *AOH* in warm-bedded regions with **(a-b, d-i)** corresponding to the GHFs **(a–h)** in Fig. 2. The blue region indicates cold-bedded areas. **(c)** Locations of specularity content, same as Fig. 3c. The blue star represents Dome C.

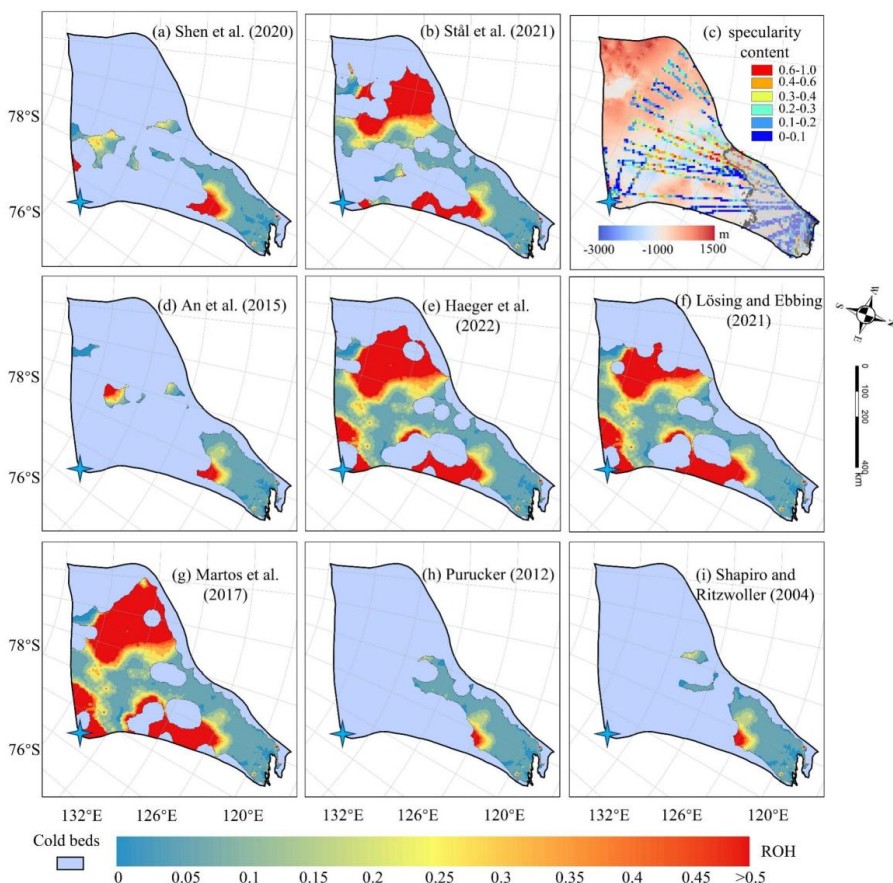

**Figure 6.** The spatial distribution of relative overheating (*ROH*) inconsistency in warm beds with (a), (b) and (d) to (i) corresponding to the GHFs (a – h) in Figure 2. The light purple mask represents the cold beds. (c) Locations of specularity content (coloured points), same as Fig. 5.

### 3.3  Evaluation of Model Inconsistency with Eight GHFs

All inconsistency indices for the simulation results using the eight GHF datasets are illustrated in Fig. 7. The overheating inconsistency associated with Purucker et al. (2012) and Shapiro and Ritzwoller (2004) GHFs is predominantly localized in fast-flowing regions. Consequently, after normalization by the surface observed ice speed, their relative rankings improve (Fig. 7). The GHFs from Purucker et al. (2012), An et al. (2015), Shapiro and Ritzwoller (2004), and Shen et al. (2020) demonstrate balanced performance with respect to both overheating and overcooling inconsistency metrics,



thereby securing the top four positions in both *ACI* and *RCI*. Their *ACI* values exhibit
similarity, ranging from 0.50 to 0.59 (Fig. 7c). In contrast, simulation result utilizing
Martos et al. (2017) GHF exhibits low *AOC* but high *AOH*. Simulation results utilizing
Stål et al. (2021) GHF show low *ROC* but relatively high *ROH*. Notably, simulation
results employing GHFs from Martos et al. (2017), Haeger et al. (2022), and Lösing
and Ebbing (2021) demonstrate comparably high *AOH* values. These four GHF
datasets—Martos et al. (2017), Stål et al. (2021), Haeger et al. (2022), and Lösing and
Ebbing (2021)—are ranked in the bottom four positions for both *ACI* and *RCI* metrics.
Furthermore, the ranking order of the eight GHFs remains consistent between *ACI* and
*RCI*.
We combine the above six metrics using the ranking of each metric from 1 to 8,
with 1 denoting the smallest inconsistency and 8 the largest. The final averaged ranking
(Fig. 7d) using the arithmetic mean of the individual metric scores,is the same as that
of *ACI* and *RCI*. Purucker et al. (2012), An et al. (2015) and Shapiro and Ritzwoller
(2004) GHFs occupy the top three positions. Following closely, Shen et al. (2020) and
Stål et al. (2021) GHFs secure the 4th and 5th positions, respectively. Martos et al.
(2017), Haeger et al. (2022) and Lösing and Ebbing (2021) GHFs are ranked as the
bottom three among the eight GHFs in Totten Glacier. The thermal state produced by
the optimal GHF result shows that warm beds predominantly cluster around the
grounding line and its upstream regions. Conversely, the inland areas of Totten largely
exhibit cold temperatures, with relatively sparse warm-bedded areas.





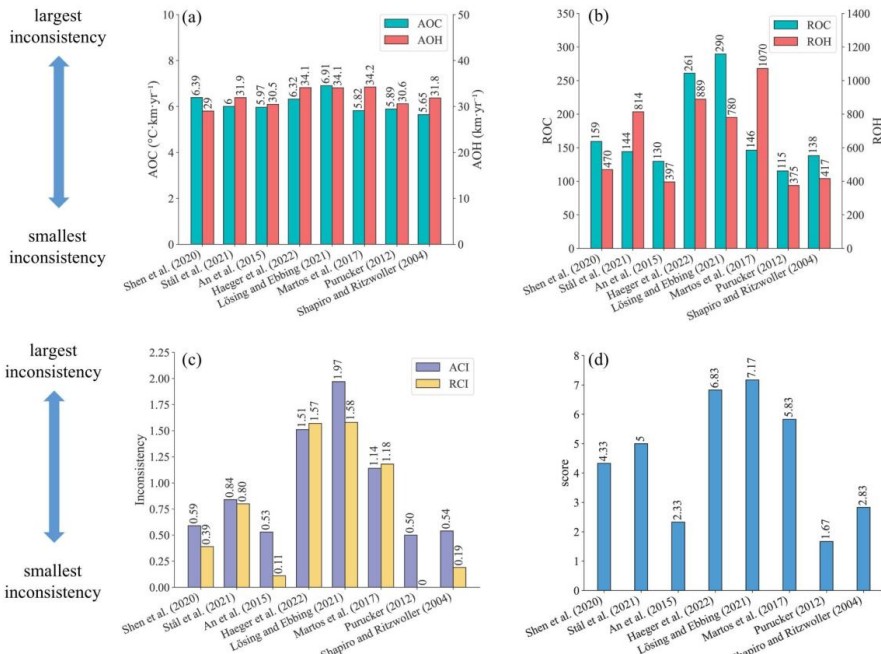

**Figure 7.** Six inconsistency indicators and the final ranking of 8 GHF datasets. **(a)** the cumulative values of *AOC* across grid points over cold bed region and *AOH* across grid points over warm bed region; **(b)** the cumulative values of *ROC* across grid points over cold bed region and *ROH* across grid points over warm bed region; **(c)** the absolute and relative combined inconsistencies, *ACI* and *RCI*; **(d)** the average of ranking scores from 1 to 8 using the six inconsistency indicators. The value of inconsistencies and scores are labeled at the top of the bars.

## 4. Discussion

### 4.1 Causes of Inconsistencies and Sources of Uncertainty

Our method evaluates the quality of an ice sheet temperature field by quantifying the inconsistency between that temperature field and the velocity field that is obtained if that temperature field is used to compute the rheology in a mechanical inversion. Because mechanical inversions use surface velocity observations as a constraint, we have developed an indirect method for using surface velocity observations to check the quality of an englacial temperature simulation. However, the mere fact that inconsistencies exist does not by itself tell us what caused those inconsistencies.

Broadly speaking, the measured inconsistencies can come from two sources: temperature or velocity. Uncertainties in any of the input datasets used to compute those





two fields can produce inconsistencies, as can simplifications in the model physics. Here, we have tested the influence of one particular boundary condition, GHF, since that field is particularly hard to constrain. Because all other inputs are kept constant, the differences in the inconsistencies that we calculated between different simulations can be attributed to the GHF fields. However, we also found that all of the models we tested had non-zero inconsistency (Fig. 3; Fig. 5). The absolute inconsistencies, *AOH* and *AOC*, had particularly small between-model variability in comparison to their mean value. This could be because none of the input GHF fields correctly captured the true GHF, but it could also indicate problems with other model inputs. For instance, the surface temperature used in Huang et al. (2024) represents the present-day climate, but the thermal structure of the ice sheet may reflect colder temperatures during the last glacial cycle. We discuss an additional experiment we performed to test the influence of uncertainty in surface temperature on our inconsistency metrics in Section 4.3 below. By contrast, surface accumulation rate should have been lower during glacial periods, which would have a warming influence on ice sheet temperatures. Uncertainties in bed topography should influence both our thermal and our mechanical models, with deeper ice being more likely to be warm, and with errors in ice thickness producing compensating errors in basal sliding in our mechanical inversion. In the study of Huang et al. (2024), BedMachine v2 was used for ice thickness and subglacial topography. However, Bedmap3 (Pritchard et al., 2025) has better-resolved mountains and smoother trough margins.

The simulation results we use from Huang et al. (2024) came from a 3D isotropic full-Stokes ice flow model. While full-Stokes is generally considered the gold standard of ice sheet mechanical modeling, the use of an isotropic rheology may not be valid in some parts of the ice sheet, such as near ice divides or at the margin of an ice stream where the history of past ice deformation creates anisotropic crystal fabric that affects the present-day mechanical properties (Martín et al., 2009; Zhao et al., 2018b; Zwinger et al., 2014). Isotropic flow laws often require the use of an "enhancement factor" for vertical shear in the lower part of the ice column, an ad hoc correction that would have a particularly large influence on our computed overcooling metrics. Thus the isotropic flow law potentially introduces errors in modelled strain rates and, hence, bias in basal sliding velocities obtained by inversion methods (Budd and Jacka, 1989; Gerber et al., 2023; Rathmann and Lilien, 2022). Simulated surface ice velocities can be influenced by other factors in addition to ice fabric; shear margins are also impacted by accumulated rupture, such as damage along a shear margin (e.g., Benn et al., 2022; Lhermitte et al., 2020; Schoof, 2004; Sun et al., 2017). Ice deposited during the last glaciation has different chemistry (especially concentrations of chloride and possibly sulphate ions) which leads to smaller crystals that develop a strong, near-vertical, single-maximum fabric (Paterson, 1991). However, ice fabric data is sparse, known




from direct observations at ice cores (Azuma and Higashi, 1985) or inferred from
specialized radar measurements (Fujita and Mae, 1994; Jordan et al., 2022), and its
impact beyond the scope of this study as we refrain from incorporating additional
observational data relying only on widely-available surface ice velocities.

Our inconsistency metrics are designed to provide bidirectional constraints,
wherein the model is penalized for both overheating and overcooling. By adopting this
bidirectional constraint framework, we aim to mitigate the risk of unidirectional
constraints leading to excessively cold or warm outcomes being deemed optimal.
However, our inconsistency metrics only provide a bidirectional constraint when
viewed in a spatially integrated sense. Locally, we only have unidirectional constraints.
This is because our overheating metrics are only computed where the bed is at the
melting point, and our overcooling metrics are only computed where the bed is below
the melting point. This makes methodological sense, as we know for sure that sliding
must only occur where the bed is warm. However, in reality it is entirely possible that
some of the areas where the modelled bed reaches the pressure melting point are still
too cold (the modelled melt rate is lower than the real melt rate), and conversely, it is
also possible that some of the areas where the modelled bed is below the pressure
melting point are still too warm (the real temperature is colder still). Our method cannot
identify these areas. Thus, our inconsistency metrics may underestimate variability in
the ice sheet thermal state: we have no way to penalize cold regions that are not cold
enough or warm regions that are not warm enough. We leave the development of these
constraints to future work.

## 4.2 Sensitivity of Inconsistencies to GHF Datasets

Comparing the GHF dataset rankings between this study and Huang et al. (2024),
we find that the top 4 and the bottom 4 are the same in the two studies, albeit with slight
variations in ranking. The lower ranking of Shen et al. (2020) in this study may be
attributed to several factors. Firstly, Huang et al. (2024) excludes areas with ice speed
exceeding 30 m a$^{-1}$ (Fig. 3c) because specularity content is an ambiguous indicator of
wet beds there. Secondly, the GHF from Shen et al. (2020) yields higher basal
temperature and also faster basal ice velocities in most of the cold bed of Totten Glacier,
hence exhibits greater overcooling inconsistency, compared with Purucker et al. (2012),
leading to a decrease in its rankings (Fig. S3). Lastly, Huang et al. (2024) primarily
relied on specularity content, while our study evaluated datasets based on
inconsistencies in the simulation results. Despite these methodological differences, both
studies identified four relatively well-performing GHF datasets for Totten Glacier,
which exhibit similar distributions of warm and cold beds when compared to the other
four datasets (Fig. 3 and Fig. 5). This similarity underscores that the warm bed is
concentrated near and upstream of the grounding line. Datasets from Stål et al. (2021),



Martos et al. (2017), Haeger et al. (2022), and Lösing and Ebbing (2021) exhibit a
tendency to overestimate GHF in central Totten Glacier.
Simulations employing GHF datasets from Stål et al. (2021), Martos et al. (2017),
Haeger et al. (2022), and Lösing and Ebbing (2021) yield more extensive warm-bedded
regions and are expected to exhibit greater overheating inconsistency. Nevertheless,
these models also exhibit relatively high overcooling inconsistency despite the limited
extent of cold-bedded regions. We quantified the discrepancies between these four GHF
datasets and the Purucker et al. (2012) GHF in terms of modelled basal velocity, basal
temperature relative to the pressure melting point, and *AOC* (Fig. S4). The Purucker et
al. (2012) GHF yields lower basal ice temperatures and slower basal velocities across
most cold-bedded regions, consequently resulting in lower *AOC* values compared to
the other four GHF datasets.

### 4.3   Implications for Ice Sheet Dynamics

There is a common area between 69°S and 72°S in the eastern part of Totten
Glacier with the largest *AOH* (Fig. 5) for all the GHFs varying from 48 to 70 mW m$^{-2}$,
which suggests that the *AOH* inconsistency is from other ice sheet properties rather than
GHF. Zhang et al. (2022) reconstructed Antarctic near-surface air temperature based on
MODIS land surface temperature measurements and in situ air temperature records
from meteorological stations from 2001 to 2018. We compared the reconstruction of
near-surface air temperature in the year 2001 (Zhang et al., 2022) and the ALBMAP v1
dataset used in Huang et al. (2024). The surface air temperature in the area with large
*AOH* from ALBMAP v1 is 0.6-3.1 °C higher than that from the reconstructed near-
surface air temperature in 2001 (Fig. 8). The MODIS-based near-surface air
temperature product shows warming in Totten Glacier from 2001 to 2018. Even so, the
surface air temperature in the area with large *AOH* from ALBMAP v1 is still higher
than that in 2018 but over a smaller area. Therefore, we infer that the large *AOH* may
be attributed to the present-day ice surface temperature derived from ALBMAP v1 in
this area being unrealistically warm. The englacial temperature will be lower than
present-day ice sheet surface temperature used in the model but warmer than the
average surface temperature during the last glacial-interglacial cycle. We lowered the
surface ice temperature in this area by 1 °C, reran the simulation, and found that *AOH*
with all the GHFs was halved (Fig. 8e).

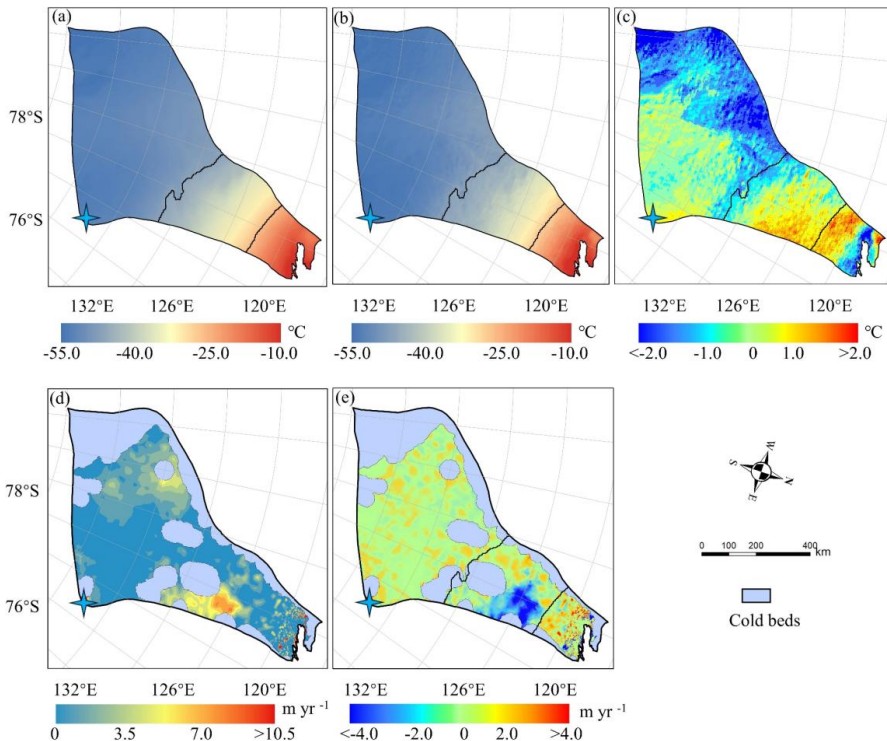

**Figure 8.** Surface ice temperature from ALBMAP v1 (a) and MODIS-based near-surface air temperature (b) in the year 2001, and their difference (c). (d) The AOH using modified surface ice temperature by reducing the temperature between the two thick black curves (contour lines of −44 ℃ and −26 ℃) in (a) by 1℃ and GHF of Martos et al. (2017). (e) The difference between the AOH using cooler surface ice temperature and the original AOH. The blue star represents Dome C.

Given that data assimilation and inverse methods are widely employed to infer basal friction coefficients in ice sheet simulations, it is essential to acknowledge the impact of the inconsistencies identified in our study on ice sheet dynamics. A cold bed is supposed to provide substantial resistance and limit basal sliding; however, if the basal temperature is overestimated, it may decrease viscosity and enhance basal sliding. This overheating inconsistency would lead to an overestimation of ice flow speeds, discharge, and the dynamic ice loss (Artemieva, 2022; Burton-Johnson et al., 2020). Similarly, under representation of warm bedding would slow ice discharge estimates, and hence potential ice sheet response to climate warming. The basal thermal regime





critically influences the stability of grounding lines and the behavior of ice streams. In
a warming climate, increases in geothermal or frictional heating can trigger basal
thawing in these areas, lowering basal friction and potentially initiating rapid grounding
line retreat—a key component of marine ice sheet instability (MISI) (Reese et al., 2023;
Ross et al., 2012). Without incorporating a self-consistent thermal model into the
inversion, projections may misrepresent the onset and extent of these dynamic
instabilities. Our findings underscore that a fully coupled inversion framework would
use not only surface velocity data but also incorporate direct or proxy observations of
basal temperature and subglacial hydrology. Such an approach would better constrain
the basal friction coefficient in a physically consistent manner, reducing the risk of
producing nonphysical states. This integration is especially critical for projections of
ice sheet evolution under climate change, as the dynamic response is sensitive to even
small changes in basal conditions.
**497    5. Conclusion**
We propose a novel and rapid method to quantify the inconsistencies between
modelled basal ice temperature and observed surface ice speed and assess the quality
of ice sheet model simulation results without using subglacial observation data.
Previously, it has been assumed that checking the quality of an ice sheet temperature
model required in situ observations, whether from ice cores or geophysical techniques
like ice penetrating radar. By using the ice temperature field to compute the rheology
structure needed for a mechanical inversion and then quantifying the inconsistency
between the inverted velocity field and the original ice temperature field, we are able
to use remotely sensed surface velocity observations as a check on the quality of
modelled englacial temperatures. Given the challenges in acquiring subglacial data, our
method can provide a streamlined and effective approach to evaluation.
We apply this method to the simulation results of Totten Glacier using a 3D full-
Stokes model with 8 different GHF datasets. Assuming the inconsistencies are mainly
due to unrealistic GHF datasets, we use the inconsistencies to assess the reliability of
those GHF datasets. We compare our GHF ranking with that by Huang et al. (2024)
which used specularity content to derive a two-sided constraint on the basal thermal
state. We find that the top 4 and the bottom 4 GHFs are the same in the two studies,
albeit with slight variations in ranking. Furthermore, we find that the simulations with
all GHF datasets underestimate the basal ice temperature in a canyon on the western
boundary of Totten Glacier, and we infer that the common high overheating
inconsistencies with all the GHF datasets in the eastern Totten Glacier between 69°S
and 72°S may be attributed to the unrealistically warm surface ice temperature used
there in the model. While we demonstrate that this approach works on simulation results
for Totten Glacier, testing of the method on other glaciers would be useful to assess if



the approach is worthwhile for revealing ambiguous conflicts in observations and
simulations.

*Data availability.* MEaSUREs BedMachine Antarctica, version 2, is available at
https://doi.org/10.5067/E1QL9HFQ7A8M (Morlighem, 2020). MEaSUREs InSAR-
Based Antarctic Ice Velocity Map, version 2, is available at
https://doi.org/10.5067/D7GK8F5J8M8R (Rignot et al., 2017). MEaSUREs Antarctic
Boundaries for IPY 2007–2009 from Satellite Radar, version 2, is available at
https://doi.org/10.5067/AXE4121732AD (Mouginot et al., 2017). ALBMAP v1 and the
GHF dataset of Shapiro and Ritzwoller (2004) are available at
https://doi.org/10.1594/PANGAEA.734145 (Le Brocq et al., 2010b). The GHF dataset
of An et al. (2015) is available at
http://www.seismolab.org/model/antarctica/lithosphere/AN1-HF.tar.gz (last access: 11
April 2023). The GHF dataset of Shen et al. (2020) is available at
https://sites.google.com/view/weisen/research-products?authuser=0 (last access: 11
April 2023). The GHF dataset of Martos (2017) is available at
https://doi.org/10.1594/PANGAEA.882503. The GHF dataset of Purucker (2012) is
available                                                                                    at
https://core2.gsfc.nasa.gov/research/purucker/heatflux_mf7_foxmaule05.txt        (last
access:11 April 2023).

*Author contributions.* LZ and JCM conceived the study. LZ, MW, and JCM designed
the methodology. JW and LZ analyzed the data and conducted visualization. JW
and LZ wrote the original draft, and all the authors revised the paper.

*Competing interests.* The contact author has declared that none of the authors has any
competing interests.

*Acknowledgements.* This work was supported by National Key Research and
Development Program of China (grant no. 2021YFB3900105) and Academy of
Finland (grant no. 355572).

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
