# Peer review of "Quantifying Temperature-sliding Inconsistency in Thermomechanical Coupling: A"

_EGUsphere, 2025_

## Referee Comment (RC1)

**Review of Wang et al. (2025) 'Quantifying Temperature-sliding Inconsistency in Thermomechanical Coupling: A Comparative Analysis of Geothermal Heat Flux Datasets at Totten Glacier'**

**Summary**

This paper presents a new set of metrics to assess the inconsistencies between modelled basal temperatures and observed surface ice velocities in ice-sheet simulations, and applies them to evaluate the likely shortcomings of eight different geothermal-heat-flux datasets in the Totten basin in East Antarctica. The paper finds that its results in this regard agree with previous work that assessed the eight datasets using radar specularity observations to determine the presence of basal water and thus whether the ice was warm or cold, which suggests it is indeed performing well in identifying problem areas.

I think the new method is a useful way to quantify the mismatches between modelled ice temperatures and observed velocities, and the paper is well-executed. I do think it would benefit from a small amount of restructuring, as presenting the application of the method and the comparison of the geothermal datasets at the same time makes it harder for the reader to understand what's going on. I also find it dissatisfying that the method cannot say anything about the source of the inconsistencies, however. Pointing out where the inconsistencies are is, of course, a useful first step, but without knowing where they're coming from, there's not necessarily much one can do with the information. That would obviously be too much for this paper though, and the authors suggest this might be the focus of future work – I would strongly encourage them to pursue it, as that would be really very useful!

Overall, I think the paper is suitable for publication subject to minor revisions to improve the clarity of the presentation.

Page and line numbers refer to those in the clean version of the submitted manuscript.

**Major Comments**
- Focus of the paper: This is a a a bit of a minor gripe, but I feel the focus of the paper is not always very clear: is it presenting the new set of metrics or is it assessing the differences between the GTF datasets? I think it would be clearer to start with one simulation where the authors show how the method works in practice, and then move on to the eight simulations comparing the GTF data. As it stands, the authors are trying to do both things at the same time and the consequent seesawing makes it harder for the reader to understand what the metrics are showing in concrete terms. This need not be a new simulation or anything – just pick one of the eight and add a short section to the results showing the fields for the metrics and explaining the interpretation, before launching in to the full-bore comparison.
- Future steps: This paper is very much a first step – yes, knowing where the inconsistencies are is good, but knowing why they're there is a lot more useful if one wants to improve one's ice-sheet model. I know the authors hint at this being a possibility for future work, but maybe they could suggest in the discussion some ways in which modellers might use the method to investigate the causes of inconsistencies? This would make the paper a lot more directly useful to the ice-sheet-modelling community, one feels.

**Minor Comments**
- p.1, l.30-35: I can't quite work out from the phrasing in the abstract for the first inconsistency if this means the GHF datasets are too cold or whether the model's too cold. Similarly, with the second one, are you saying the model is overheating compared to the GHF datasets, or that the datasets themselves are too hot? Or both? Obviously, it's explained later, but best to not confuse people with the first thing they'll read! Some rephrasing to make it clearer what's going on here would be beneficial.

---

## Author Comment (AC1)

Referee's comments are in blue, our reply in black, quotes in the revised manuscript in red.

**Review of Wang et al. (2025) 'Quantifying Temperature-sliding Inconsistency in Thermomechanical Coupling: A Comparative Analysis of Geothermal Heat Flux Datasets at Totten Glacier'**

**Summary**

This paper presents a new set of metrics to assess the inconsistencies between modelled basal temperatures and observed surface ice velocities in ice-sheet simulations, and applies them to evaluate the likely shortcomings of eight different geothermal-heat-flux datasets in the Totten basin in East Antarctica. The paper finds that its results in this regard agree with previous work that assessed the eight datasets using radar specularity observations to determine the presence of basal water and thus whether the ice was warm or cold, which suggests it is indeed performing well in identifying problem areas.

I think the new method is a useful way to quantify the mismatches between modelled ice temperatures and observed velocities, and the paper is well-executed. I do think it would benefit from a small amount of restructuring, as presenting the application of the method and the comparison of the geothermal datasets at the same time makes it harder for the reader to understand what's going on. I also find it dissatisfying that the method cannot say anything about the source of the inconsistencies, however. Pointing out where the inconsistencies are is, of course, a useful first step, but without knowing where they're coming from, there's not necessarily much one can do with the information. That would obviously be too much for this paper though, and the authors suggest this might be the focus of future work – I would strongly encourage them to pursue it, as that would be really very useful!

Overall, I think the paper is suitable for publication subject to minor revisions to improve the clarity of the presentation.

Reply: Thank you for your encouraging comments.

Page and line numbers refer to those in the clean version of the submitted manuscript.

**Major Comments**

• Focus of the paper: This is a a bit of a minor gripe, but I feel the focus of the paper is not always very clear: is it presenting the new set of metrics or is it assessing the differences between the GTF datasets? I think it would be clearer to start with one simulation where the authors show how the method works in practice, and then move on to the eight simulations comparing the GTF data. As it stands, the authors are trying to do both things at the same time and the consequent seesawing makes it harder for the reader to understand what the metrics are showing in concrete terms. This need not be a new simulation or anything – just pick one of the eight and add a

short section to the results showing the fields for the metrics and explaining the interpretation, before launching in to the full bore comparison.

Reply: Thank you for your comments. We add a section in the revision. We show the spatial fields of the inconsistencies metrics (Section 2.1) for the modelled result in Huang et al. (2024) with Martos et al. (2017) GHF as an example, and provide an interpretation, before conducting a comprehensive comparative analysis for the result with 8 GHF datasets.

Here is the section we add:

**3.1 Spatial Distribution of Inconsistencies with one GHF dataset**

In this section, we show the spatial fields of the inconsistencies metrics (Section 2.1) for the modelled result in Huang et al. (2024) with Martos et al. (2017) GHF as an example, and provide an interpretation before conducting a comprehensive comparative analysis for the result with 8 GHF datasets.

Figure 3. Spatial distribution of modelled basal ice temperature using the Martos et al. (2017) GHF. (a), modelled basal ice speed (b), AOC (c), ROC (d) inconsistencies in modelled cold-bedded regions, and AOH (e) and ROH (f) inconsistencies in modelled warm-bedded regions. The colormap in (c) and (d) is on logarithmic scale. The pink region in (c) and (d) represents modelled thawed bed, while the blue region in (e) and (f) indicates frozen-bedded areas.

The modelled result based on the Martos et al. (2017) GHF reveals extensive

regions of thawed bed with limited areas of frozen bed. The frozen bed is predominantly located in the southern corner of the study domain, where the modelled basal ice speed approaches zero, consistent with cold basal ice temperature. Consequently, the AOC inconsistency at this marginal zone is negligible (Fig. 3). Along the western margin of Totten Glacier, basal ice temperature remains below the pressure melting point, albeit approaching it. However, localized regions exhibit high basal velocities of several tens of meters per year, contradicting the presence of a frozen bed and resulting in large AOC inconsistencies.

On the other hand, large AOH values are observed between 69°S and 71°S in the eastern Totten Glacier region, where the simulated surface ice speed exceeds observational data by >5 m a-1 (Fig. 3e). In this area, the modelled basal ice temperature reaches the pressure melting point, with the modelled basal ice speed at approximately 0.05 m a-1. Basal friction inversion failed to reproduce observed surface ice speed due to the model's overestimation of ice temperature and softness. This pronounced velocity mismatch highlights a fundamental inconsistency in the eastern glacier region, likely originating from discrepancies in the input datasets. Regions of high ROH and ROC values coincide with areas of relatively high AOH and ROC, particularly where the observed surface velocities are slow, as per their formulations.

• Future steps: This paper is very much a first step – yes, knowing where the inconsistencies are is good, but knowing why they're there is a lot more useful if one wants to improve one's ice-sheet model. I know the authors hint at this being a possibility for future work, but maybe they could suggest in the discussion some ways in which modellers might use the method to investigate the causes of inconsistencies? This would make the paper a lot more directly useful to the ice-sheet modelling community, one feels.

Reply: Evaluating the inconsistencies is like doing basal friction inversion. It tells us where is the misfit, but it alone cannot tell us why there is a misfit. Anyhow, there is a procedure that could be worth checking. So we make a list in the discussion that one could check in the revision.

Evaluating the inconsistencies reveals where mismatches occur but not why they arise. The primary factors to investigate are surface temperature, GHF, accumulation rate, and ice thickness, representing the most critical boundary conditions. Furthermore, integrating multiple sources of prior knowledge can help constrain model parameters:

- 1. High-resolution radar measurements: Check availability of ice thickness data along flight lines to validate geometric boundary conditions.
- 2. Paleoclimate context: Historical climate reconstructions indicate significantly colder surface temperatures during glacial periods compared to present-day conditions, with correspondingly lower accumulation rates. These paleo-temperature conditions likely induced a long-term thermal memory within the ice column, potentially contributing to observed discrepancies between modeled and measured basal properties.

Therefore, we recommend systematically evaluating:

- (1) Spatial distribution of radar-derived ice thickness measurements;
- (2) Temporal consistency of surface temperature boundary conditions;
- (3) Sensitivity of model results to GHF;
- (4) Accumulation rate reconstructions during key climatic periods.

This multi-faceted approach helps isolate the causes of inconsistencies in ice sheet simulations.

**Minor Comments**

• p.1, l.30-35: I can't quite work out from the phrasing in the abstract for the first inconsistency if this means the GHF datasets are too cold or whether the model's too cold. Similarly, with the second one, are you saying the model is overheating compared to the GHF datasets, or that the datasets themselves are too hot? Or both? Obviously, it's explained later, but best to not confuse people with the first thing they'll read! Some rephrasing to make it clearer what's going on here would be beneficial.

Reply: We rephrase these sentences

"Examples of the method utility are 1. an inconsistency characterizing overcooling with all GHFs near the western boundary of Totten Glacier between 70°S-72°S, where there is a bedrock canyon and fast surface ice velocities, which suggests that GHF is low in all published datasets; 2. an overheating inconsistency in the eastern Totten Glacier with all GHFs that leads to overestimation of ice temperature due, in this case, to an unrealistically warm surface temperature."

to

"Examples of the method utility are 1. an overcooling inconsistency with all GHFs near the western boundary of Totten Glacier between 70°S-72°S, where there is a bedrock canyon and fast surface ice velocities, suggesting that all GHFs are low; 2. an overheating inconsistency in the eastern Totten Glacier with all GHFs suggesting overestimation of ice temperature, in this case, due to an unrealistically warm surface temperature."

---

## Author Comment (AC2)

Referee's comments are in blue, our reply in black, quotes in the revised manuscript in red.

In this study, Wang et al. compare modelled velocities derived from a full-Stokes thermomechanical model (from a previous study, Huang et al., 2024) with surface velocity observations from MEaSUREs. I do not think this study is a novel contribution to the field, and found the manuscript confusing, so I do not recommend it for publication.

**Major comments**

In a preceding study, Huang et al. (2024) conduct full-Stokes thermomechanical simulation forced by 8 different geothermal heat flux models. The strength of this study is the comparison of the model results with an independent constraint – the radar specularity content of the bed. To make this comparison, Huang et al. perform an inversion for the basal parameters, based on modelled vs. surface velocities. So in this study, Wang et al. are calculating metrics based on the residuals of the modelled vs surface velocities, and are thus evaluating the performance of the Huang et al. inversions. While it seems reasonable that better performing inversions reflect more accurate GHF map, a range of other factors can be at play. For instance, the authors note the possible influence of anisotropic viscosity. Poorly-performing inversions could also be the result of uncertainties in the form of the basal sliding law, or in the basal topography model. Wang et al. present a ranking of GHF maps that is similar but not identical to that of Huang et al. It is unclear why Wang et al. think that this ranking is more accurate, especially given the circularity of the velocity residual argument mentioned above. It is also unclear that the Wang et al study presents results which are novel, relative to the Huang et al. study.

Reply: Thanks for the referee's comments. There is a misunderstanding from the referee regarding our method. In Huang et al. (2024), modelled surface velocity is compared with the observed over the whole domain in the inversion for basal parameters for each GHF data. The referee said "So in this study, Wang et al. are calculating metrics based on the residuals of the modelled vs surface velocities, and are thus evaluating the performance of the Huang et al. inversions." While it is true that our overheating metrics are based on (a subset of) the velocity residuals from the inversion, our overcooling metrics are based on the basal sliding velocity from the inversion, which does not enter into the residual of a mechanical inversion at all, since the observational constraints for a mechanical inversion are surface velocities. Moreover, a mechanical inversion does not take into account the physical plausibility of the sliding result it produces. It is not circular reasoning to compare two different parts of a model to each other; rather, it is a check of internal consistency, or lack thereof. A mechanical inversion may well be able to fit the surface velocity observations equally well when forced with many different models of the ice sheet thermal structure and rheology; however, if some of those models require high sliding velocities in cold-based regions, then they should be downweighted in comparison to models that show a good agreement between basal temperature and velocity.

We understand how this can appear to be getting something from nothing, so to speak, since our method does not require any additional observations beyond the surface velocities used in the mechanical inversion. However, it is wrong to say that we do not have any "independent constraints" in this method. The "independent constraint" that we are using here is not an observation, but rather the a priori physical understandings that: 1) rapid sliding requires warm basal temperatures and subglacial water, and, 2) reducing the basal slip coefficient cannot prevent the ice from flowing by internal shear deformation. The inconsistency metrics that we developed in this paper are an attempt to quantify and rank the extent to which these basic (and uncontroversial) physical understandings are violated.

Of course, the reviewer is correct that other factors beyond GHF can cause inconsistencies between model velocities and temperatures. We discuss some of those factors in the discussion section of the manuscript. However, we used the same forcing fields for all of our models except for GHF, so it is fair to attribute differences in the inconsistency metrics to the only factor that varied between the models, which was GHF.

As for the referee's comment, "Wang et al. present a ranking of GHF maps that is similar but not identical to that of Huang et al. It is unclear why Wang et al. think that this ranking is more accurate,", we do not claim that our ranking is more accurate than theirs. The purpose of demonstrating this method in the same domain as used by Huang et al is so that we can validate the method against their results based on radar specularity measurements, which is useful before applying our method to other domains where such measurements are not available. However, it would be unlikely for both methods to produce exactly the same ranking. The two methods both have their own sources of uncertainty and their own sets of assumptions and arbitrary parameter choices. We are not claiming that either ranking is better in any sort of absolute sense, only that they both seem to be measuring roughly the same thing.

Wang et al. justify their study by saying that they are introducing a new method – this is misleading. They are instead introducing new terminology for "metrics" based on velocity residuals, which is a common practice in the field of glaciological inversion. One metric masks the thawed bed, another metric masks the frozen bed, but both metrics are used in the final evaluation – so it is unclear why the masking was necessary to start with. I don't understand why you need "bidirectional constraints" when you could take the root-mean-square error. Even if they were presenting a new method, they do not provide any tests to show that this method improves upon existing ones. Finally, the terminology of these metrics is confusing, and I found did not provide me with a better physical understanding of why some models performed

**better than others (although this last point can be rectified with clearer language).**

Reply: As we discussed in our reply above, there seems to be a confusion here: the overheating metrics use the surface velocities, and can thus be thought of as a subset of the inversion residual, but the overcooling metrics use the basal velocities, and thus they are not in any sense a subset of the inversion residual. The "new method" that we are introducing is the practice of quantifying the internal (in)consistency between a sliding inversion and the rheology structure used to force that inversion. The specific metrics that we use to quantify this inconsistency could be changed, for example by using a squared error term as the reviewer suggests instead of the linear error terms that we used, but the general practice of caring about and quantifying the inconsistency between a sliding inversion and the temperature/rheology field used as an input to that inversion is new, and that is what we mean when we describe our contribution as a new method.

We were concerned about the need for "bidirectional constraints" in the manuscript because we wanted to avoid a situation in which we only penalized models for being too cold or too warm, and thus ended up with a ranking that favored the most extreme GHF maps rather than those which accurately balanced warm and cold conditions. The reviewer does not seem to understand why we were concerned about this, because they assumed that all of our metrics were computed from the surface velocities, and thus that they could be easily combined into a single RMS metric just as in the cost function for the sliding inversion. However, our overcooling metrics are computed by comparing basal velocity with basal temperature, and basal velocity does not appear in the inversion cost function. This is the metric that is easiest to explain and understand: areas that are cold-based should not be sliding. But if we only used overcooling metrics, then our final ranking would be biased towards the hottest GHF maps, regardless of whether those were the most accurate. This is why we also needed overheating metrics to obtain an unbiased result. As we explained in the methods section, our overheating metrics are built around the physical understanding that a sliding inversion cannot prevent the ice from flowing by internal shear deformation. The inversion cannot reduce the sliding velocity below zero, and since it does not adjust the near-basal rheology of the ice, it is powerless to reduce the residual with the observations if that residual is caused by internal deformation. Thus if the basal ice is warmer and softer than it should be, we should expect the inversion to be biased towards positive residuals (meaning the model surface velocity is faster than the observed) in areas of warm bed. This is why we use a subset of the inversion residual for our overheating metrics.

We acknowledge that it might be preferable to use one metric or constraint over the whole domain. But we, nor anyone else to our knowledge, has proposed and applied the same metric on both warm bed and cold bed. For instance, we can use the temperature difference between ice temperature and pressure melting point in the metric for the cold bed. But we cannot use it on the warm bed, as the ice temperature at the warm bed equals the pressure melting point.

Finally, we do not claim that our method improves upon existing ones, which we assume you refer to Huang et al. (2024). As we discussed above, Huang et al. (2024) used radar specularity observations to evaluate the model result. But we do not use any englacial or subglacial data in this study. The significance of this work is that the metrics we proposed provide an effective approach to evaluate modelling results and identify problem areas without using any englacial or subglacial measurement data.

**Minor comments**

In Figure 3 & 4, I think it would be clearer to change the term "warm bed" to "thawed bed". And in Figure 5, "cold bed" to "frozen bed".

Reply: Okay, we change "warm bed" to "thawed bed", and "cold bed" to "frozen bed" in Fig. 3, 4 and 5.

The authors should note where they obtain their velocity observations within the main text, and maybe plot them.

Reply: Thanks for your suggestions. In the main text, we write

Huang et al. (2024) used the present-day surface ice temperature (Le Brocq et al., 2010), observed surface velocity from MEaSUREs InSAR-Based Antarctic Ice Velocity Map, version 2 (Rignot et al., 2017) and ice sheet topography data from BedMachine Antarctica, version 2 (Morlighem et al., 2020).

We also add a plot of surface velocity observation in Figure 1.

Figure 1. (a) Geographic location of Totten Glacier (red outline) in Antarctica; (b) bed elevation of Totten Glacier, the purple curve represents the grounding line; (c) observed surface velocity.

I think the authors should add a section describing the model methodology, so that the reader can understand this paper without having to read Huang et al.

Reply: Thanks for your suggestion. We add the following subsection in the revision:

**2.2 Methodology in Huang et al. (2024)**

Huang et al. (2024) employed thermo-mechanical coupled simulations using eight GHF datasets to investigate the steady-state thermal regime of Totten Glacier. The methodology involved two interconnected modeling components:

- 1. Forward Modeling: An enhanced shallow-ice approximation model integrated with a subglacial hydrology module was utilized to simulate englacial temperature profiles.
- 2. Inverse Problem: A full-Stokes ice flow model was applied to resolve basal friction coefficients through inverse analysis, to minimize the misfit between simulated and observed velocities while simultaneously generating velocity predictions.

A feedback loop was established: the velocity outputs from the inverse model were used to refine key parameters in the forward model - specifically constraining the basal slip ratio, rheological properties, and shape functions. This bidirectional coupling process underwent multiple iterations to achieve convergent steady-state solutions. Huang et al. (2024) further used specularity content as a critical constraint to differentiate localized wet versus dry basal conditions. They compared modeled basal thermal states with different GHFs and observational returns, and thus evaluate the reliability of GHF datasets.

---

## Author Response (AR1)

Editor's comments are in blue, our reply in black, quotes in the revised manuscript in red.

Dear Junshun Wang and co-authors,

Your manuscript has received two constructive reviews that reflect a wide range of opinions. This divergence highlights both the potential and the current limitations of the work. On one hand, the study introduces an approach that could provide new insights into geothermal heat flux variability in Antarctica by quantifying internal inconsistencies in ice sheet models. This is a potentially valuable contribution to the modeling community, especially in regions where direct observations are sparse.

On the other hand, several major issues were identified, most notably by Reviewer #2. A key concern is the clarity of the manuscript's novelty and its distinction from previous work, particularly Huang et al. (2024). While you have provided thoughtful and detailed responses, it remains important to explicitly clarify the unique contribution of this study, both in methodology and in application, relative to earlier work. I strongly encourage you to address this directly, perhaps through a dedicated paragraph or subsection within the revised manuscript.

In summary, while the manuscript shows promise, it requires substantial revision to address concerns related to novelty, clarity, and interpretation. I therefore encourage you to submit a thoroughly revised version. Once received, I will seek reevaluation by the reviewers to determine whether the revisions meet the standards for publication in The Cryosphere.

Best regards,
Cheng Gong

Reply: Thanks for the editor's comments. To address concerns related to novelty, clarity, and interpretation, we carefully improved the revision.

We clarity the inconsistencies in Section 2.1 as below:
The inconsistencies defined in this study are essentially between a sliding inversion and the temperature/rheology field used as an input to that inversion. More specifically, the inconsistencies are between modelled basal sliding (which is tuned to match the observed fast surface velocity during the inversion) and modelled frozen bed, and between observed slow surface velocity (which is most likely indicative of a non-slip basal condition) and modelled thawed bed.

We add two short sections to show the difference in methodology between this study and Huang et al. (2024), and clarify the novelty and the unique contribution of this study.

**2.2 Methodology in Huang et al. (2024)**

Huang et al. (2024) employed thermo-mechanical coupled simulations using eight GHF datasets to investigate the steady-state thermal regime of Totten Glacier. The methodology comprised two interconnected modeling components:

1. Forward Modeling: An enhanced shallow-ice approximation model integrated with a subglacial hydrology module was utilized to simulate englacial temperature profiles.

2. Inverse Problem: A full-Stokes ice flow model was applied to resolve basal friction coefficients through inverse analysis, to minimize the misfit between simulated and observed velocities while simultaneously generating velocity predictions.

A feedback loop was then established: the velocity outputs from the inverse model were used to refine key parameters in the forward model - specifically constraining the basal slip ratio, rheological properties, and shape functions. This bidirectional coupling process underwent multiple iterations to achieve convergent steady-state solutions.

Huang et al. (2024) utilized radar specularity content data to differentiate localized wet (thawed) versus dry (frozen) basal conditions and used this data as a two-sided constraint on the basal thermal state. They compared modeled basal thermal states derived from different GHFs to evaluate the reliability of the GHF datasets.

**2.3 Distinction from Huang et al. (2024)**

In Huang et al. (2024), modelled surface velocity velocities are compared with observations over the whole domain during the inversion for basal parameters for each GHF dataset. Here, surface velocities act as the observational constraints for the mechanical inversion.

Although the overheating metrics here use the surface velocities and can thus be considered a subset of the inversion residual, our overcooling metrics are based on the basal sliding velocity derived from the inversion, which is not part of the mechanical inversion's residual. A mechanical inversion does not take into account the physical plausibility of the sliding result it produces. Therefore, it is not circular reasoning to compare two different parts of a model to each other; rather, it is a check of internal consistency, or lack thereof. A mechanical inversion may fit the surface velocity observations equally well when forced with many different models of the ice sheet thermal structure and rheology; however, if some models require high sliding velocities in frozen-based regions, then they should be downweighted in comparison to models that show a good agreement between basal temperature and velocity.

The method here does not require any additional observations beyond the surface velocities used in the mechanical inversion. However, there are "independent constraints" in the method here, which are not observations, but rather the a priori physical understandings that: 1) rapid sliding requires warm basal temperatures and subglacial water; 2) reducing the basal slip coefficient cannot prevent the ice from flowing by internal shear deformation. The inconsistency metrics developed in this paper are an attempt to quantify and rank the extent to which these basic (and uncontroversial) physical understandings are violated.

We changed "warm bed" to "thawed bed", and "cold bed" to "frozen bed" in the text according to the comment by Reviewer 2.

We also add a section as below according to the comment by Reviewer 1. We show the spatial fields of the inconsistencies metrics (Section 2.1) for the modelled result in Huang et al. (2024) with Martos et al. (2017) GHF as an example, and provide an interpretation, before conducting a comprehensive comparative analysis for the result with 8 GHF datasets.

**3.2 Spatial Distribution of Inconsistencies with one GHF dataset**

In this section, we show the spatial fields of the inconsistency metrics (Section 2.1) for the modelled result in Huang et al. (2024), using Martos et al. (2017) GHF as an example. This example illustrates the interpretation process before conducting a comprehensive comparative analysis for the result with 8 GHF datasets.

[Figure]

Figure 3. Spatial distribution of modelled basal ice temperature (a), modelled basal ice speed (b), *AOC* (c), *ROC* (d) inconsistencies in modelled frozen-bedded regions, and *AOH* (e) and *ROH* (f) inconsistencies in modelled thawed-bedded regions associated with Martos et al. (2017) GHF. The colormap in (c) and (d) is on logarithmic scale. The pink region in (c) and (d) represents modelled thawed bed, while the blue region in (e) and (f) indicates frozen-bedded areas.

The modelled result based on the Martos et al. (2017) GHF reveals extensive regions of thawed bed with limited areas of frozen bed. The frozen bed is predominantly located in the southern corner of the study domain, where the modelled basal ice speed

approaches zero, consistent with cold basal ice temperature. Consequently, the AOC inconsistency at this marginal zone is negligible (Fig. 3). Along the western margin of Totten Glacier, basal ice temperature remains below the pressure melting point, albeit approaching it. However, localized regions exhibit high basal velocities of several tens of meters per year, contradicting the presence of a frozen bed and resulting in large AOC inconsistencies.

Conversely, large AOH values are observed between 69°S and 71°S in the eastern Totten Glacier region, where the simulated surface ice speed exceeds observational data by >5 m a$^{-1}$ (Fig. 3e). In this area, the modelled basal ice temperature reaches the pressure melting point, with the modelled basal ice speed at approximately 0.05 m a$^{-1}$. Basal friction inversion failed to reproduce observed surface ice speed due to the model's overestimation of ice temperature and softness. This pronounced velocity mismatch highlights a fundamental inconsistency in the eastern glacier region, likely originating from discrepancies in the input datasets. Regions of high ROH and ROC values coincide with areas of relatively high AOH and AOC, particularly where the observed surface velocities are slow, as per their formulations.

We also add a paragraph in the discussion to show the things one could check to isolate the causes of inconsistencies in application.

While evaluating inconsistencies highlights the spatial distribution of mismatches, it does not inherently elucidate their underlying causes. The primary factors to investigate are surface temperature, GHF, accumulation rate, and ice thickness, representing the most critical boundary conditions. Furthermore, integrating multiple sources of prior knowledge can help constrain model parameters:
1. High-resolution radar measurements: The availability of ice thickness data along flight lines should be assessed to validate geometric boundary conditions.
2. Paleoclimate context: Historical climate reconstructions indicate significantly colder surface temperatures during glacial periods compared to present-day conditions, with correspondingly lower accumulation rates. These paleo-temperature conditions likely induced a long-term thermal memory within the ice column, potentially contributing to observed discrepancies between modeled and measured basal properties.

Therefore, we recommend a systematic evaluation of: (1) The spatial distribution of radar-derived ice thickness measurements; (2) The temporal consistency of surface temperature boundary conditions; (3) The sensitivity of model results to GHF variations; (4) Accumulation rate reconstructions during key climatic periods. This multi-faceted approach helps isolate the causes of inconsistencies in ice sheet simulations.

We hope these edits can address concerns related to novelty, clarity, and interpretation of this study.

---

## Referee Report (RR1)

**Summary:** The paper investigates inconsistencies in ice sheet models when the basal thermal state is inferred from ice surface velocities. The authors derive six metrics with which they quantify these inconsistencies, apply their method to Totten Glacier, Antarctica, using eight geothermal heat flux models, and compare their results with radar specularity. The authors further use the discovered inconsistencies to rank the eight geothermal heat flux models in their reliability. I commence the authors for finding this novel approach and I see the added value in having an evaluation method that can be applied rapidly and with relatively little input data. However, there are inconsistencies with terminology, methodology, and clarity of the manuscript so that revisions ae required.

I have two major comments and a number of minor comments that are outlined below:

**Major**:

1. The manuscript would benefit from a clearer structure. As the paper combines a multitude of modelled and observed data, as well as several different metrics, it is hard to follow. The results section in particular is very hard to follow as it switches between GHF products, absolute and relative metrics as well as overcooling and overheating. I suggest the following:
   a. Clearly define your input datasets and metrics in the methodology section. Use different subheadings for a) Definition of Metrics B) Normalization and ranking. The section could also benefit from a table that shows all input datasets.
   b. While the maps are useful to assess the spatial distribution of different metrics, I would also add a table that shows the key differences for each metric and GHF.
   c. Structure the results section by metric with individual subheadings e.g. absolute inconsistencies, relative inconsistencies, comparison.
   d. Section 4.3. should be more on the caveats as it mostly discusses the influence of near-surface air temperatures on your results. I would change the subheading to e.g. "Impact of Input datasets", and rather than calling it an additional experiment, state that a caveat of your study is that it is influenced by the input data.
2. At various times in the manuscript, the authors label datasets and values as unrealistic, gold standard or otherwise. These labels are unscientific and should be removed (see more detailed comments below for occurrences that I have spotted).

**Minor**:

Throughout the manuscript the authors switch between surface velocity and surface speed. For consistency, it should be one or the other.

Line 85: Insert "the" before basal friction coefficient

Line 125 – 132: This paragraph could go into the introduction as you define inconsistencies there. I would, however, remove the sentence in Lines 92-94 and rephrase the sentence in Lines 125 – 126 to the following: "*For this study we define inconsistencies as differences between modelled frozen bed and modelled basal sliding (which is tuned to match the observed fast surface velocity during the inversion), and between modelled warm bed and observed slow surface velocity. The inconsistencies originate from multiple causes, including uncertainties in GHF, surface ice temperature, ice sheet geometry, bed topography, surface velocity, ice density and incomplete ice flow mechanics.*"

Lines 155 – 156: This sentence is unnecessary. Consider removing.

Lines 197 – 199: This sentence can be incorporated into the next one e.g. *"We obtain three absolute inconsistencies (AOH, AOC, ACI) and three relative inconsistencies (ROH, ROC, RCI), with which we can comprehensively analyze …"*

Line 230: It is not quite clear from the map where 71°S is as the map only shows 76°S and 78°S. The map should either show the coordinates referred to in the text or the area should be highlighted somehow.

Line 234: Being colder than what? I assume than the other GHF products. Consider adding *"… than the other four GHF products.".*

Line 240: The canyon is not really apparent in Figure 1b - Consider adding an outline.

Lines 244 – 245: This should go further up in the methods section where you define AOH and AOC.

Lines 246 – 247: Again, it is not clear where the area referred to in the text is located as the coordinates in the map do not correspond.

Line 254 – 256: This sentence refers to ice flow and references Figure 3, which does not show ice flow. I would suggest referring to Figure S2 for ice velocity and Figure 3 at the end of the sentence.

Line 261 – 263: I would expect the spatial distribution of the two metrics to be different as they are derived differently. Consider removing the sentence as I think it is not providing any additional information. If it stays in the manuscript, it should be absolute "overheating" inconsistencies in Line 262.

Line 263: Mention how to find Dome C in the figure (e.g. Dome C (Blue Star, Figure 4).

Figures 4 – 6: Keep the wording consistent. Where the color scale is logarithmic, say that instead of "non-linear". Consider changing the color for the Dome C marker – It is very hard to see. Also, if Dome C is marked in all figures, it should also be referred to in the figure caption.

Lines 282 – 284: Repetition from Lines 216 – 218. Consider removing.

Line 287: Again, the coordinates are meaningless if the map doesn't reflect them.

Figure 7 a,b: Not the best colors for colorblindness. Consider changing (check here: https://colorbrewer2.org/)

Lines 345 – 347: Repetition from Lines 125 – 132. Consider removing.

Line 349: "check" sounds a bit informal. Maybe use "assess".

Lines 361 – 362: This is a very strong statement, and I don't think you can say that unless you have a product that captures GHF correctly or a citation to back it up. Consider removing or at least toning it down.

Lines 367 –368: In contrast to what - Not quite clear why this sentence follows the section on uncertainty metrics. Maybe it was supposed to go after Line 365. Also, it should be "in contrast" not "by".

Line 376: Considered "gold standard" by who? Either add a citation or refrain from using ratings.

Line 395: Insert "is" between "impact" and "beyond".

Line 405: Remove "we know for sure".

Figure 8 – caption: Change "thick black curves" to "black lines".

Line 480: Remove "the" before "dynamic ice loss".

Lines 481 – 482: Rephrase. Currently, the sentence doesn't make any sense.

Lines 482 – 483: This sentence needs a citation.

Line 494: Change "under climate change" to "under future climate change scenarios".

Line 501: Change "checking" to "assessing".

Line 507: This is the first time you talk about englacial temperature. I assume you mean basal temperature.

Line 509: Which simulation results? Add a citation or refer to a specific simulation.

Line 511: Again, this is a strong assumption unless you know what is realistic. Maybe use "due to differences in".

Lines 518 – 519: See comments above on coordinates.

Line 519: Find a different word unless you can prove that is not realistic

Line 520: Remove "there".

**Supplementary material**

Figure S1 – caption: Add abbreviation for pressure melting point. It is otherwise not clear what PMP in the legend stands for.

Figure S4: The labels should include the year of the publication as is present in all other figures (e.g. Purucker (2012) – Lösing and Ebbing (2021))

As you are citing in the figure captions, you should add a reference list.

---

## Referee Report (RR2)

**Review of Wang et al. (2025) 'Quantifying Temperature-sliding Inconsistency in Thermomechanical Coupling: A Comparative Analysis of Geothermal Heat Flux Datasets at Totten Glacier'**

**Summary**
This paper presents a new set of metrics to assess the inconsistencies between modelled basal temperatures and observed surface ice velocities in ice-sheet simulations, and applies them to evaluate the likely shortcomings of eight different geothermal-heat-flux datasets in the Totten basin in East Antarctica. The paper finds that its results in this regard agree with previous work that assessed the eight datasets using radar specularity observations to determine the presence of basal water and thus whether the ice was warm or cold, which suggests it is indeed performing well in identifying problem areas, and validates the method as a way of assessing the consistency of simulation results.

I reviewed the first version of this paper and had some relatively minor comments of a structural nature. I am pleased to see that the authors have addressed these and I think the clarity and flow of the paper are now much better. I have only a couple of small further comments related to the new material added in response to the first round of review, but, otherwise, I think the paper is ready for publication at this stage. So, minor revisions, but no more than that!

Page and line numbers refer to those in the clean version of the submitted manuscript.

**Major Comments**
- None

**Minor Comments**
- Section 2.2-2.3: I know why these sections are here, having read all the reviews, but, if one hasn't, these both intrude rather awkwardly into the paper, seemingly for no reason. I think the easiest way of solving this might be to just add a sentence to the start of Section 2.2 saying something like 'We validate our work in this study by comparing our ranking of GHF datasets to the observationally constrained ranking of Huang et al. (2024). For readers not familiar with this paper, we provide here a brief summary of their method and, in the next section, clarify the distinction between their paper and the present study.' Then at least readers will understand the point of these sections and the paper will flow a bit more naturally.
- Section 4.2-4.3: I might suggest swapping the order of these two sections. 4.3 follows on naturally from the discussion of possible mismatch causes in 4.1, so having the unrelated relatively technical section on sensitivity that is 4.2 in-between them feels a little odd.

---

## Author Response (AR2)

**Response to the Editor & Reviewers**

**Response to the Editor**

Editor's comments are in blue, our reply in black.

Dear Junshun Wang and co-authors,
Thank you for your careful revisions and for engaging constructively with the reviewers' feedback. In the second round of review, both referees provided positive evaluations and acknowledged the scientific contribution and improvements made to your manuscript.

While the overall reception has been favorable, Referee #3 has raised a number of constructive comments aimed at further improving the clarity and structure of the manuscript.
I therefore invite you to submit a revised version that responds carefully to all remaining comments, particularly those from Referee #3.

Best regards,
Cheng Gong
Editor, The Cryosphere

Dear editor,
We sincerely thank you for your handling of our manuscript and for the positive assessment provided by the referees in the second round of review. We are pleased to submit the revised version of our manuscript.

We have carefully addressed all remaining comments. In particular, we have made significant efforts to further improve the clarity and structure of the manuscript to ensure the presentation meets the high standards of the journal.

Below, we provide a point-by-point response to the specific comments raised.

Sincerely,
Liyun Zhao (on behalf of all co-authors)

**Response to Referee #1**

Referee's comments are in blue, our reply in black, quotes in the revised manuscript in red.

**Review of Wang et al. (2025) 'Quantifying Temperature-sliding Inconsistency in Thermomechanical Coupling: A Comparative Analysis of Geothermal Heat Flux Datasets at Totten Glacier'**

**Summary**

This paper presents a new set of metrics to assess the inconsistencies between modelled basal temperatures and observed surface ice velocities in ice-sheet simulations, and applies them to evaluate the likely shortcomings of eight different geothermal-heat-flux datasets in the Totten basin in East Antarctica. The paper finds that its results in this regard agree with previous work that assessed the eight datasets using radar specularity observations to determine the presence of basal water and thus whether the ice was warm or cold, which suggests it is indeed performing well in identifying problem areas, and validates the method as a way of assessing the consistency of simulation results.

I reviewed the first version of this paper and had some relatively minor comments of a structural nature. I am pleased to see that the authors have addressed these and I think the clarity and flow of the paper are now much better. I have only a couple of small further comments related to the new material added in response to the first round of review, but, otherwise, I think the paper is ready for publication at this stage. So, minor revisions, but no more than that!

Reply: Thank you for your encouraging comments.

Page and line numbers refer to those in the clean version of the submitted manuscript.

**Major Comments**
• None.

**Minor Comments**
• Section 2.2-2.3: I know why these sections are here, having read all the reviews, but, if one hasn't, these both intrude rather awkwardly into the paper, seemingly for no reason. I think the easiest way of solving this might be to just add a sentence to the start of Section 2.2 saying something like 'We validate our work in this study by comparing our ranking of GHF datasets to the observationally constrained ranking of Huang et al. (2024). For readers not familiar with this paper, we provide here a brief summary of their method and, in the next section, clarify the distinction between their paper and the present study.' Then at least readers will understand the point of these sections and the paper will flow a bit more naturally.

Reply: Thank you for this insightful observation regarding the structure and flow of the

manuscript. We have added an introductory sentence at the beginning of Section 2.2 to explicitly state the motivation for including these sections.

In this study, we validate our method by comparing our ranking of GHF datasets to the observationally constrained ranking established by Huang et al. (2024). For readers not familiar with this paper, we provide here a brief summary of their method and, in the next section, clarify the distinction between their paper and the present study.

• Section 4.2-4.3: I might suggest swapping the order of these two sections. 4.3 follows on naturally from the discussion of possible mismatch causes in 4.1, so having the unrelated relatively technical section on sensitivity that is 4.2 in-between them feels a little odd.

Reply: Thanks for your suggestion. We agree that the interpretative discussion presented in Section 4.3 follows naturally from the analysis of potential mismatch causes in section 4.1. Section 4.2 (Sensitivity Analysis) is technically independent. We also think section 4.3 has a natural transition toward the Conclusion section, so we would like to keep section 4.3, and swap the order of the original Section 4.1 and 4.2.

Therefore, the new structure is as follows:
1. **Section 4.1:** Sensitivity of Inconsistencies to GHF Datasets (former 4.2)
2. **Section 4.2:** Causes of Inconsistencies and Sources of Uncertainty (former 4.1)
3. **Section 4.3:** Implications for Ice Sheet Dynamics

**Response to Referee #3**

Referee's comments are in blue, our reply in black, quotes in the revised manuscript in red.

**Summary:** The paper investigates inconsistencies in ice sheet models when the basal thermal state is inferred from ice surface velocities. The authors derive six metrics with which they quantify these inconsistencies, apply their method to Totten Glacier, Antarctica, using eight geothermal heat flux models, and compare their results with radar specularity. The authors further use the discovered inconsistencies to rank the eight geothermal heat flux models in their reliability. I commence the authors for finding this novel approach and I see the added value in having an evaluation method that can be applied rapidly and with relatively little input data. However, there are inconsistencies with terminology, methodology, and clarity of the manuscript so that revisions are required.

Reply: Thank you for your encouraging comments. For the Line numbers that the referee mentioned, we assume the referee read the original version rather than the earlier version of revision we submitted. But it does not prevent us from identifying the places that need revision.

I have two major comments and a number of minor comments that are outlined below: **Major**:

1. The manuscript would benefit from a clearer structure. As the paper combines a multitude of modelled and observed data, as well as several different metrics, it is hard to follow. The results section in particular is very hard to follow as it switches between GHF products, absolute and relative metrics as well as overcooling and overheating. I suggest the following:

a. Clearly define your input datasets and metrics in the methodology section. Use different subheadings for a) Definition of Metrics B) Normalization and ranking. The section could also benefit from a table that shows all input datasets.

b. While the maps are useful to assess the spatial distribution of different metrics, I would also add a table that shows the key differences for each metric and GHF.

c. Structure the results section by metric with individual subheadings e.g. absolute inconsistencies, relative inconsistencies, comparison.

d. Section 4.3. should be more on the caveats as it mostly discusses the influence of near-surface air temperatures on your results. I would change the subheading to e.g. "Impact of Input datasets", and rather than calling it an additional experiment, state that a caveat of your study is that it is influenced by the input data.

Reply: We appreciate your detailed and constructive advice on restructuring the manuscript. We have fully adopted your suggestions to enhance clarity and readability. Our specific revisions are detailed below:

a. **Methodology Structure:** Following your suggestion, we have reorganized the Methodology section and added a summary table to clearly define the input

datasets and metrics.

We introduced specific subheadings: "2.1.1 Definition of Metrics" and "2.1.2 Normalization and Ranking".

The input datasets are described in detail in Section 3.1 of the manuscript. In addition, we have added a summary table listing all input datasets. To keep the main manuscript concise, this table is provided as Table S1 in the Supplementary Material, and referenced in the methodology section.

Table S1. Summary of input datasets used in ice sheet model.

| Input variables | Datset name | Reference |
|---|---|---|
| Surface ice velocity | MEaSUREs InSAR-Based Antarctic Ice Velocity Map, version 2 | Morlighem et al. (2020) |
| Surface elevation, bed elevation and ice thickness | MEaSUREs BedMachine Antarctica, version 2 | Rignot et al. (2017) |
| Surface temperature | ALBMAP v1 | Le Brocq et al. (2010) |
| | Antarctic_T2m_reconstruction_ 2001-2018 | Zhang et al. (2022) |
| GHF maps | — | Shen et al. (2020) |
| | | Stål et al. (2021) |
| | | An et al. (2015) |
| | | Haeger et al. (2022) |
| | | Lösing and Ebbing (2021) |
| | | Martos et al. (2017) |
| | | Purucker (2012) |
| | | Shapiro and Ritzwoller (2004) |
| Specularity content | ICECAP basal interface specularity content | Dow et al. (2019) |

b. **Summary Table in Results:** To complement the spatial maps, we have added a summary table (**Table 1**) in the Results section. This table highlights the key differences for each metric and GHF product, allowing for a more direct quantitative comparison.

Table 1. Summary of inconsistency metrics for different GHF maps.

| GHF maps | AOC ($°C km yr^{-1}$) | AOH ($km yr^{-1}$) | ROC ($°C$) | ROH | ACI | RCI |
|---|---|---|---|---|---|---|
| Shen et al. (2020) | 6.39 | 29 | 159 | 470 | 0.59 | 0.39 |
| Stål et al. (2021) | 6 | 31.9 | 144 | 814 | 0.84 | 0.8 |

| | | | | | | |
|---|---|---|---|---|---|---|
| An et al. (2015) | 5.97 | 30.5 | 130 | 397 | 0.53 | 0.11 |
| Haeger et al. (2022) | 6.32 | 34.1 | 126 | 889 | 1.51 | 1.57 |
| Lösing and Ebbing (2021) | 6.91 | 34.1 | 290 | 780 | 1.97 | 1.58 |
| Martos et al. (2017) | 5.82 | 34.2 | 146 | 1072 | 1.14 | 1.18 |
| Purucker (2012) | 5.89 | 30.6 | 115 | 375 | 0.5 | 0 |
| Shapiro and Ritzwoller (2004) | 5.65 | 31.8 | 138 | 417 | 0.54 | 0.19 |

c. **Structure of Results section:** We have restructured the Results section to align with the specific metrics. The section is now organized under the following subheadings: "3.3.1 Overcooling Inconsistency on Frozen Beds" and "3.3.2 Overheating Inconsistency on Thawed Beds".

d. **Framing of Section 4.3:** We agree that the discussion on near-surface air temperatures serves more as a caveat regarding input data sensitivity than a standalone experiment. We retitled this section as "Impact of Input Datasets".

2. At various times in the manuscript, the authors label datasets and values as unrealistic, gold standard or otherwise. These labels are unscientific and should be removed (see more detailed comments below for occurrences that I have spotted).

Reply: We agree with you that terms such as "unrealistic", "gold standard", and similar labels imply subjective judgement and are therefore not appropriate in a scientific context. Following your comment, we carefully reviewed the entire manuscript and replaced these subjective labels with more objective and precise terms.

We have revised the relevant sentence in **Section 4.2** (Line 495 – 499) to: While full-Stokes is generally considered as an ice sheet model with the most complete physical processes to date, the use of an isotropic rheology may not be valid in some parts of the ice sheet, such as near ice divides or at the margin of an ice stream where the history of past ice deformation creates anisotropic crystal fabric that affects the present-day mechanical properties.

In addition, we have removed all instances of "unrealistic" from the manuscript and rewritten the corresponding sentences to be more objective.
Example 1:
Original: Assuming the inconsistencies are mainly due to **unrealistic** GHF datasets, we use the inconsistencies to assess the reliability of those GHF datasets.
Revised: Assuming the inconsistencies are mainly due to quality issues of GHF datasets, we use the inconsistencies to assess the reliability of those GHF datasets.
Example 2:
Original: An overheating inconsistency in the eastern Totten Glacier with all GHFs suggesting overestimation of ice temperature due, in this case, to an **unrealistically** warm surface temperature.
Revised: An overheating inconsistency in the eastern Totten Glacier with all GHFs

suggesting overestimation of ice temperature due, in this case, to **a warm bias** in the surface temperature.

**Minor**:

Throughout the manuscript the authors switch between surface velocity and surface speed. For consistency, it should be one or the other.

Reply: To ensure consistency, we use "surface velocity" throughout the revision. In equations or comparisons involving only scalar values, we use the term "velocity magnitude".

Line 85: Insert "the" before basal friction coefficient.

Reply: We have added it.

Line 125 – 132: This paragraph could go into the introduction as you define inconsistencies there. I would, however, remove the sentence in Lines 92-94 and rephrase the sentence in Lines 125 – 126 to the following: "*For this study we define inconsistencies as differences between modelled frozen bed and modelled basal sliding (which is tuned to match the observed fast surface velocity during the inversion), and between modelled warm bed and observed slow surface velocity. The inconsistencies originate from multiple causes, including uncertainties in GHF, surface ice temperature, ice sheet geometry, bed topography, surface velocity, ice density and incomplete ice flow mechanics.*"

Reply: We appreciate your suggestion to define the "inconsistencies" earlier in the Introduction. Following your suggestion, we have removed the sentence "which we refer to as inconsistencies in this study" (Lines 92-94) in the Introduction, rephrased the sentences in Lines 125-132 as below, and moved them to the Introduction: "For this study, we define the inconsistencies as differences between a sliding inversion and the temperature/rheology field used as an input to that inversion. More specifically, the inconsistencies are between modelled basal sliding (which is tuned to match the observed fast surface velocity during the inversion) and modelled frozen bed, and between observed slow surface velocity (which is most likely indicative of a non-slip basal condition) and modelled thawed bed. The inconsistencies originate from multiple causes, including uncertainties in GHF, surface ice temperature, ice sheet geometry, bed topography, surface velocity, ice density and incomplete ice flow mechanics."

Lines 155 – 156: This sentence is unnecessary. Consider removing.

Reply: We have removed it.

Lines 197 – 199: This sentence can be incorporated into the next one e.g. *"We obtain three absolute inconsistencies (AOH, AOC, ACI) and three relative inconsistencies (ROH, ROC, RCI), with which we can comprehensively analyze ..."*

Reply: We have revised the manuscript and incorporated this sentence into the following one, as you suggested.

"Therefore, we obtain three absolute inconsistencies (AOH, AOC, ACI) and three

relative inconsistencies (ROH, ROC, RCI), with which we can comprehensively analyze the temperature-sliding inconsistency in the inversion results of ice sheet model."

Line 230: It is not quite clear from the map where 71°S is as the map only shows 76°S and 78°S. The map should either show the coordinates referred to in the text or the area should be highlighted somehow.

Reply: To address this, we have revised all figures containing maps in the manuscript. We have added clear coordinate labels (latitude and longitude) to the axes of these figures to ensuring that locations such as 71°S are easily identifiable. An example of the updated figures is provided below.

[Figure]

**Updated Figure 6.** Spatial distribution of *AOH* in thawed-bedded regions with **(a-b, d-i)** corresponding to the GHFs **(a–h)** in Fig. 2. The blue region indicates frozen-bedded areas. **(c)** Locations of specularity content, same as Fig. 4c. The white star represents Dome C.

Line 234: Being colder than what? I assume than the other GHF products. Consider adding *"... than the other four GHF products."*.

Reply: Thank you for identifying this ambiguity. We have added the suggested phrase to clarify the comparison.

"All 8 GHF datasets produce low basal ice temperatures in the inland southwest, with Purucker et al. (2012), Shapiro and Ritzwoller (2004), Shen et al. (2020) and Lösing and Ebbing (2021) being colder than the other four GHF products."

Line 240: The canyon is not really apparent in Figure 1b - Consider adding an outline.

Reply: Thank you for pointing this out. We agree that the canyon was not sufficiently clear in the original version of Figure 1b. We considered adding an outline (contours) to highlight the canyon; however, we found that this resulted in visual clutter and compromised the overall readability of the map. Instead, we have optimized the color scale (colormap) to enhance the contrast in this specific range. Consequently, the subglacial canyon is now clearly visible in the revised figure.

[Figure]

**Updated Figure 1. (a)** Geographic location of Totten Glacier (red outline) in Antarctica; **(b)** bed elevation of Totten Glacier, the purple curve represents the grounding line; **(c)** observed surface velocity.

Lines 244 – 245: This should go further up in the methods section where you define AOH and AOC.

Reply: We assume the referee means this sentence: "We calculate the absolute inconsistencies, AOH, in the warm bed, and AOC in the cold bed." Hence, we guess the referee read the original version rather than the earlier version of revision we submitted. The sentence in Line 244-245 of the original version is Line 320-321 in the earlier revision. We removed this sentence. We also reorganized the method section according to another referee's comments. So we do not need this sentence in the revision.

Lines 246 – 247: Again, it is not clear where the area referred to in the text is located as the coordinates in the map do not correspond.

Reply: Thank you for pointing this out. To improve clarity, we have added coordinate labels directly to the map, which now clearly indicate the location of the area referred to in the text.

Line 254 – 256: This sentence refers to ice flow and references Figure 3, which does

not show ice flow. I would suggest referring to Figure S2 for ice velocity and Figure 3 at the end of the sentence.
Reply: We added a citation to Figure S2.

Line 261 – 263: I would expect the spatial distribution of the two metrics to be different as they are derived differently. Consider removing the sentence as I think it is not providing any additional information. If it stays in the manuscript, it should be absolute "overheating" inconsistencies in Line 262.
Reply: We have removed this sentence.

Line 263: Mention how to find Dome C in the figure (e.g. Dome C (Blue Star, Figure 4).
Reply: We have added the text to help readers locate Dome C in the figure.
The largest value of ROC across most GHF occurs at Dome C (white star in Figure 5), where the observed surface ice velocity magnitudeis close to zero (Fig. 1c).

Figures 4 – 6: Keep the wording consistent. Where the color scale is logarithmic, say that instead of "non-linear". Consider changing the color for the Dome C marker – It is very hard to see. Also, if Dome C is marked in all figures, it should also be referred to in the figure caption.
Reply: Thank you for these helpful suggestions. We have ensured consistent wording across Figures 4–6 and now explicitly describe the color scale as logarithmic where applicable. The color of the Dome C marker has been changed to white to improve visibility. Dome C is marked in all figures, and referred to in the figure captions.

Lines 282 – 284: Repetition from Lines 216 – 218. Consider removing.
Reply: We have removed the sentence.

Line 287: Again, the coordinates are meaningless if the map doesn't reflect them.
Reply: Thank you for pointing this out. We have added coordinate labels to the map, which now clearly indicate the location of the area referred to in the text.

Figure 7 a,b: Not the best colors for colorblindness. Consider changing (check here)
Reply: We have redrawn the figure using a colorblind-safe palette (specifically, a distinct blue and vermilion pair adapted from the Okabe-Ito palette) to ensure distinct contrast for all readers.

[Figure]

**Updated Figure 8.** Six inconsistency indicators and the final ranking of 8 GHF datasets. **(a)** the absolute overcooling and overheating inconsistencies, AOC and AOH; **(b)** the relative overcooling and overheating inconsistencies, ROC and ROH; **(c)** the absolute and relative combined inconsistencies, ACI and RCI; **(d)** the average of ranking scores from 1 to 8 using the six inconsistency indicators. The value of inconsistencies and scores are labeled at the top of the bars.

Lines 345 – 347: Repetition from Lines 125 – 132. Consider removing.
Reply: We have removed this sentence.

Line 349: "check" sounds a bit informal. Maybe use "assess".
Reply: We have replaced the word "check" with more appropriate scientific terms such as "assess" or "evaluate" throughout the manuscript.

Lines 361 – 362: This is a very strong statement, and I don't think you can say that unless you have a product that captures GHF correctly or a citation to back it up. Consider removing or at least toning it down.
Reply: We agree with your assessment. Since ground truth GHF data is unavailable, claiming that none of the products captured the "true" GHF is indeed too definitive. We have toned it down, and rephrase this sentence
"This could be because none of the input GHF fields correctly captured the true GHF, but it could also indicate problems with other model inputs."
to

"This could be related to uncertainties or limitations in the input GHF fields, but it may also indicate sensitivities to other model inputs."

Lines 367 –368: In contrast to what - Not quite clear why this sentence follows the section on uncertainty metrics. Maybe it was supposed to go after Line 365. Also, it should be "in contrast" not "by".

Reply: We change this sentence to

"While the cooler surface temperatures during the glacial period exerted a cooling effect on ice sheet temperature, lower surface accumulation rates over the same period induced a warming effect."

Line 376: Considered "gold standard" by who? Either add a citation or refrain from using ratings.

Reply: We have removed the term "gold standard" throughout the manuscript. We change it to

"While full-Stokes is generally considered as an ice sheet model with the most complete physical processes to date, ..."

Line 395: Insert "is" between "impact" and "beyond".

Reply: Done.

Line 405: Remove "we know for sure".

Reply: Done. We change this sentence to "This makes methodological sense, as sliding is generally expected to occur where the bed is thawed."

Figure 8 – caption: Change "thick black curves" to "black lines".

Reply: We have revised this.

Line 480: Remove "the" before "dynamic ice loss".

Reply: Done.

Lines 481 – 482: Rephrase. Currently, the sentence doesn't make any sense.

Reply: We have rephrased the sentence to improve grammatical correctness and clarity regarding physical implications.

"Similarly, underrepresentation of thawed bed conditions will lead to an underestimation of ice discharge and, consequently, an underestimation of ice sheet's response to climate warming."

Lines 482 – 483: This sentence needs a citation.

Reply: We have added two references as below:

"The basal thermal regime critically influences the stability of grounding lines and the behavior of ice streams (Dawson et al., 2022; Robel et al., 2014)."

Line 494: Change "under climate change" to "under future climate change scenarios".

Reply: Done.

Reply: Done.

Reply: We changed "englacial" to "basal".

Reply: We modified this sentence as "We apply this method to evaluate the steady-state simulation results of Totten Glacier presented by Huang et al. (2024), which were derived using a 3D full-Stokes model with 8 different GHF datasets."

Reply: We agree. We have removed the word "unrealistic". We change it to
"Assuming the inconsistencies are mainly due to the quality issues of GHF datasets, ..."

Reply: We have added coordinate labels to the map figure, which now clearly indicates the location of the area referred to in the text.

Reply: We have replaced this "unrealistically warm surface ice temperature" with "a warm bias in the prescribed surface ice temperature".

Reply: We have removed it.

**Supplementary material**
Figure S1 – caption: Add abbreviation for pressure melting point. It is otherwise not clear what PMP in the legend stands for.
Reply: We have added "The abbreviation PMP stands for pressure melting point."

Figure S4: The labels should include the year of the publication as is present in all other figures (e.g. Purucker (2012) – Lösing and Ebbing (2021))
Reply: We have updated the labels in Figure S4 to include the year of publication.

As you are citing in the figure captions, you should add a reference list.
Reply: We have added a reference list in Supplementary material.